# Exploring Deep Learning Parameter Space with a-GPS: Approximate Gaussian Proposal Sampler

## Abstract

To trust the predictions provided by deep neural networks we need to quantify the uncertainty. This can be done with Bayesian neural networks. However, they require a trade-off between exactness and effectiveness. This paper introduces a new sampling framework: Adaptive Proposal Sampling (APS). APS is a mode seeking sampler that adapts the proposal to match a posterior mode. When modes overlap, APS will adapt to a new mode if it draws a sample that belongs to a new mode. A variant of APS is the approximate Gaussian Proposal Sampler (a-GPS). We show that it becomes a perfect sampler if it has the same score function as the posterior. With a warm-start of a pretrained model, combined with stochastic gradients it scales up to deep learning. Results show that a-GPS 1) proposes samples that are proportional to a mode, 2) explores multi-modal landscapes, 3) has fast computations, 4) scales to big data. Immediate results suggest that this framework may be a step towards having both exactness and effectiveness.

## 1 Introduction

Deep learning has demonstrated remarkable advancements in safety-critical domains such as driving (Kendall & Gal, 2017; Mukhoti & Gal, 2019) and health prognosis (Kleppe et al., 2022). The realization of the potential of deep learning in high-risk situations hinges on our ability to place trust in the predictions generated by these models (Kendall & Gal, 2017). This trust, in turn, demands a nuanced understanding of prediction uncertainty. As models grow in capability, there is a growing focus on researching post-hoc methods for converting pretrained models into quantifiable uncertainty models (Izmailov et al., 2019; Maddox et al., 2019; Daxberger et al., 2021; Kristiadi et al., 2020; Jospin et al., 2022).

Existing models face limitations, constrained either by their Gaussian posterior approximation (Kristiadi et al., 2020) or inefficient exploration (Welling & Teh, 2011) resulting in a low effective sample size Nemeth & Fearnhead (2021). This paper introduces a novel sampler designed to swiftly generate a posterior approximation for any model. This sampler has the flexibility to be multimodal, while ensuring a high effective sample size within modes.

### 1.1 Related Work

Markov chain Monte Carlo (MCMC) methods use a pretrained model as a warm-start to generate new models. However, classical MCMC demands substantial resources, as the Metropolis-Hastings (MH) correction relies on the entire dataset (Metropolis et al., 1953; Hastings, 1970). This computational inefficiency has encouraged the adoption of stochastic gradient MCMC (SG-MCMC) (Welling & Teh, 2011; Ma et al., 2015; Nemeth & Fearnhead, 2021) and other approximations of the MH correction (Zhang et al., 2020a; Bardenet et al., 2017; Zhang et al., 2020b).

Stochastic gradient Langevin dynamics (SGLD) (Welling & Teh, 2011) offers fast computations at the expense of slow exploration (Gal, 2016; Girolami & Calderhead, 2011; M et al., 2017). The demand for a computationally fast sampler, disregarding explorative efficiency, has motivated the application of stochastic gradient descent as an SG-MCMC method (SGD-MC) (M et al., 2017). On the other hand Hamiltonian Monte Carlo (HMC) achieves higher effective sample size (Neal, 1996a)

but at an increased computational cost. Recent work by Izmailov et al. (2021) deployed full-gradient HMC on deep learning, finding its performance comparable to the stochastic gradient version (Chen et al., 2014). However, SG-MCMC methods rely on inefficient proposal dynamics that generates few effective samples per epoch, even from a warm-start, making it expensive for deep learning problems. There is a need for a framework addressing these challenges by self-adjusting towards a *perfect sampler*, meaning that it must produce independent samples from the true distribution, obviating the need for an MH correction.

Various frameworks aim to quantify uncertainty in pretrained models. Monte Carlo dropout (MC-dropout) (Gal & Ghahramani, 2016) randomly drops neurons during training and inference, requires an architecture with dropout layers. There is need for a model-agnostic method. The Laplace approximation (LA) (Daxberger et al., 2021) estimates a Gaussian posterior shape using the pretrained weights as the mean, typically limited to the last layer. There is a need for an approach that can be applied to all layers. Stochastic weight averaging (SWA) (Izmailov et al., 2019) and SWA-Gaussian (SWAG) (Maddox et al., 2019) use the trajectory of SGD-MC inside a mode around the pretrained weights. These methods restrict their posterior approximation to a Gaussian. There is a need for a multi-modal posterior. MC-dropout, LA and SWA suffer limitations in terms of inexactness and slow exploration, restrictions to a unimodal posterior, reliance on specific model architectures, or computational inefficiency during inference. There is a need for a framework that overcomes these limitations.

## 1.2 OUR CONTRIBUTIONS

In this paper, we introduce a novel sampling framework with distinct features.

- First, the method is self-adjusting towards a **perfect sampler**, meaning that it produces independent samples from the true distribution, obviating the need for an MH correction.
- Second, unlike MC-Dropout, our approach is **model-agnostic**, ensuring versatility across various models.
- Third, the proposed method **can be applied to large models**, thus overcoming the notable memory weakness of LA.
- Fourth, the proposed method **supports a multi-modal posterior**, thus overcoming limitations in methods like LA, SWA, and SWAG.
- Fifth, the proposed model demonstrates **higher efficiency** compared to SG-MCMC, SWA, and SWAG. It has **linear inference time** with respect to the number of samples, in contrast to SWA and SWAG. The method provides efficient exploration and computation, crucial for dealing with large models and datasets.

Our method surpasses the limitations encountered by SG-MCMC, MC-dropout, LA, and SWA, providing a solution that mitigates computational demands during training/sampling, addressing issues of inexactness and slow exploration, eliminating restrictions to a unimodal posterior, operating independently of specific model architectures, and ensuring computational efficiency during inference, thereby fulfilling the crucial need for a comprehensive framework. These contributions collectively position our framework as a robust and versatile tool for Bayesian inference tasks.

## 2 ADAPTIVE PROPOSAL SAMPLING

The Adaptive Proposal Sampler (APS) is a framework designed to dynamically adjust the parameters of a proposal distribution, aiming to replicate the local shape of the posterior distribution $f$. APS operates under the assumption that modes share the same score function ($\nabla \log f$) as the selected proposal distribution $q$. This mode-seeking sampler adapts $\nabla \log q$ to align with the score function of a posterior mode. In the case of overlapping modes, APS adjusts to a new mode when it draws a sample belonging to a new mode. The primary objective of APS is to generate samples that faithfully represent the true distribution, steering clear of inefficient random walk behavior. We posit that if the sampler aligns with the score function, a Metropolis-Hastings correction becomes unnecessary.

A specific instance of APS is the Gaussian Proposal Sampler. We demonstrate that it converges to a perfect sampler when it shares the same score function as the target. A perfect sampler $q$ generates

a sample $\theta$ with the exact probability of $f(\theta)$ (Propp & Wilson, 1996). We advocate for the use of the Gaussian Proposal Sampler due to its effectiveness in handling multi-modal distributions. In Appendix I we also propose parameter updates for a Beta and Gamma Proposal Sampler.

## 2.1 THE GAUSSIAN PROPOSAL SAMPLER (GPS)

Assuming the posterior has a density function

$$f(\theta) = c \exp g(\theta) \propto \mathcal{N}(\mu, \sigma^2). \tag{1}$$

To sequentially sample from the posterior, we assume access to $\theta_t$ and the score function $\nabla \log f(\theta_t)$, the derivative of $\log f(\theta_t)$ with respect to $\theta_t$. As $\nabla \log f(\theta)$ is independent of the scalar $c$, we propose to approximate $f(\theta)$ with a Gaussian distribution $q = \mathcal{N}(\mu_{t+1}, \sigma^2_{t+1})$. We have

$$\nabla \log f(\theta) \approx \nabla \log q(\theta), \tag{2a}$$

$$= -\frac{(\theta - \mu_{t+1})}{\sigma^2_{t+1}}. \tag{2b}$$

Setting these equal at $\theta = \theta_t$, we express the conditional update for $\mu_{t+1}$ as

$$\mu_{t+1} \mid \sigma^2_{t+1} = \theta_t + \sigma^2_{t+1} \times \nabla \log f(\theta_t). \tag{3}$$

For the update of $\sigma^2_{t+1}$, we note that

$$\nabla^2 \log f(\theta) \approx \nabla^2 \log q(\theta) \tag{4a}$$

$$= \frac{-1}{\sigma^2_{t+1}}. \tag{4b}$$

An approximation of $\nabla^2 \log f(\theta)$ can be expressed as

$$\nabla^2 \log f(\theta) \approx \frac{\nabla \log f(\theta) - \nabla \log f(\mu_t)}{\theta - \mu_t}. \tag{5}$$

Setting this equal at $\theta = \theta_t$, we get

$$\nabla^2 \log q(\theta_t) = \nabla^2 \log f(\theta_t), \tag{6a}$$

$$\frac{-1}{\sigma^2_{t+1}} = \frac{\nabla \log f(\theta_t) - \nabla \log f(\mu_t)}{\theta_t - \mu_t}. \tag{6b}$$

Thus, the conditional update for $\sigma^2_{t+1}$ is

$$\sigma^2_{t+1} \mid \mu_t = \left| \frac{\theta_t - \mu_t}{\nabla \log f(\theta_t)} \right|. \tag{7}$$

These conditional parameter updates from Equations (3) and (7) provide a $q$ that approximates the score function of $f$. Refer to Appendix B for proof that $q$ converges to match the score function of any $f \propto \mathcal{N}(\mu, \sigma^2)$ and becomes a *perfect sampler*. For $\theta \in \mathbf{R}^d$, the parameters are considered independent, making the proposal a diagonal multivariate Gaussian $q = \mathcal{N}_d(\mu_{t+1}, \sigma^2_{t+1}\mathbf{I}_{d \times d})$, where $\mu_{t+1} \in \mathbf{R}^d$ and $\sigma^2_{t+1} \in \mathbf{R}^d$. When $f$ is an arbitrary density, the MH correction is typically needed to ensure exactness in the sampler dynamics, so we provide an exact GPS algorithm in Appendix C.

For stability and efficiency, we introduce an upper variance limit $\lambda$. This hyperparameter limits the potential risk of overestimating the underlying variance around saddle points or other flat areas, or to ensure that samples remain sufficiently close to high density area (see Appendix E, for more details). Although it can limit the exploration, if modes are far apart, to counter this the posterior can be tempered (Neal, 1996b) to flatten the modes.

**Algorithm 1** Approximate Gaussian Proposal Sampler

**Input:** Specify $\lambda, \mu_0, \sigma_0$
$\theta_1 \sim \mathcal{N}(\mu_0, \sigma_0^2)$

**for** $t = 1$ **to** $T$ samples **do**

$\sigma_{t+1}^2 \leftarrow \left| \frac{\theta_t - \mu_t}{\nabla \log f(\theta_t)} \right|$

$\mu_{t+1} \leftarrow \theta_t + \sigma_{t+1}^2 \times \nabla \log f(\theta_t)$

$\sqrt{\sigma_{t+1}^2} = \min(\sqrt{\sigma_{t+1}^2}, \lambda)$

$\theta^{t+1} \sim \mathcal{N}(\mu_{t+1}, \sigma_{t+1}^2)$

**end for**

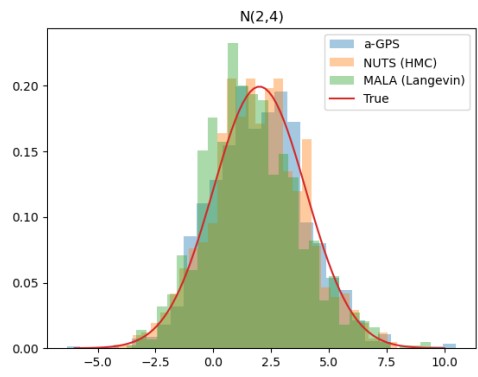

Figure 1: Histograms of 1000 samples from a single run with various methods for the target distribution $\mathcal{N}(2, 4)$

### 2.1.1 APPROXIMATE GAUSSIAN PROPOSAL SAMPLER (A-GPS)

We argue that we can ignore the Metropolis-Hastings (MH) correction without suffering from inefficient sampling and slow exploration. This MH-free version of GPS is referred to as approximate GPS (a-GPS). Consider the case of Equation (1); as the Gaussian proposal adapts its parameters

$$q(\theta) \to d \exp g(\theta) \propto f(\theta), \tag{8}$$

the MH acceptance rate $\alpha \to 1$ (see the Appendix B for proof). In the more general case of $N$ modes when

$$f(\theta) = \sum_{i=1}^{N} f_i(\theta), \tag{9}$$

where $f_i$ is a mode as:

$$f_i(\theta) = c_i \exp g_i(\theta) \propto \mathcal{N}(\mu_i, \sigma_i^2), \tag{10}$$

then for a mode $f_i$, we still have that if

$$q(\theta) \to d \exp g(\theta) \propto f_i(\theta), \tag{11}$$

then the MH acceptance rate for the specific mode $\alpha_i \to 1$, where $\alpha_i$ is defined as:

$$\alpha_i(\theta, \theta*) = \min\{1, \frac{f_i(\theta*)q(\theta|\theta*)}{f_i(\theta)q(\theta*|\theta)}\}. \tag{12}$$

To motivate the use of a mode-specific MH instead of global MH, consider that in deep learning it is intractable to the explore all modes, as there are millions (Jospin et al., 2022). The predictive performance benefits from exploring some different modes (Abe et al., 2022; Fort et al., 2021; 2020). Most of the modes are equally good (Keskar et al., 2017; LeCun et al., 2015). The modes are assumed to be Gaussian (Daxberger et al., 2021; Maddox et al., 2019; Izmailov et al., 2019). The differences between true gradient and stochastic gradient MCMC in deep learning are negligible (Izmailov et al., 2021). Thus, it is not possible to explore all the modes in the true distribution, though we want samples from some different modes. Since the modes can be approximated by a Gaussian, we have the case of Equation (9), and since the modes are equally good the estimation of mixing parameters $c_i$ are not important. What is important is to get samples from some modes, which we can achieve with a mode specific MH. However, if the modes can be approximated with a Gaussian, then with GPS the mode-specific MH $\alpha_i \to 1$. Therefore we can use GPS without the MH mode correction, which is the approximate GPS (pseudo code for a-GPS in Algorithm 1). Finally, since stochastic gradients are sufficient, we propose to use a-GPS with stochastic gradients for deep learning.

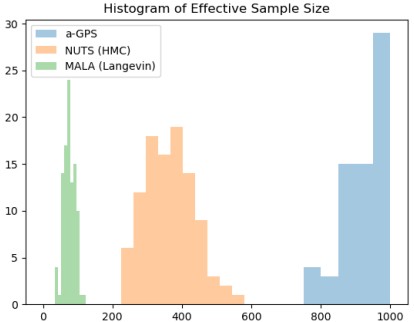 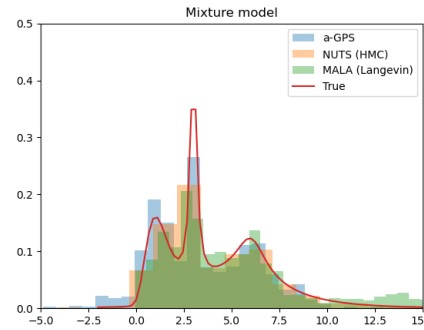

Figure 2: **Left:** Histogram of the effective sample size on the unimodal example, replicated 100 times. **Right:** The samples of retrieved from one run with a-GPS, NUTS and MALA when the true distribution is a multi-modal mixture model.

## 3 RESULTS AND DISCUSSION

The following results present a collection of experiments conducted on simulated data, CI-FAR10/100 (Krizhevsky, 2009), and ImageNet (Deng et al., 2009). The employed methods include NUTS (Hoffman & Gelman, 2014), Langevin dynamics Roberts & Rosenthal (2001), Laplace approximation (Daxberger et al., 2021), MC-dropout (Gal & Ghahramani, 2016) (Note: VGG-16 is the only architecture with dropout nodes), stochastic weight averaging (SWA) (Izmailov et al., 2019), stochastic weight averaging Gaussian (SWAG) (Maddox et al., 2019), stochastic gradient descent MCMC (SGD-MC) (M et al., 2017), and our method a-GPS. When addressing both SWA and SWAG, we may use the notation SWA(G).

### 3.1 SIMULATED EXPERIMENTS

In this section, we compare a-GPS (ours) against the successful No-U-Turn sampler (NUTS) (Hoffman & Gelman, 2014). NUTS is a version of Hamiltonian Monte Carlo and is regarded as the gold standard baseline. Additionally, we include the Metropolis-adjusted Langevin algorithm (MALA). We test the sampler on a Gaussian target and a mixture of non-Gaussian distributions.

In the first example we have a Gaussian mode $\mathcal{N}(2, 4)$. For our method we set the initial values as $\lambda = 10$, $\mu_0 = 0$ and $\sigma_0 = 1$ and our initialisation is $\theta_1 \sim \mathcal{N}(\mu_0, \sigma_0^2)$. Then a-GPS generates 1000 samples, without burn-in. We found that a-GPS converges towards a perfect sampler in just a few steps so a dedicated burn-in period was unnecessary. For NUTS we used PyRo's implementation (Bingham et al., 2019), and their automatic tuning parameter schedule, the burn-in time was set to the standard of 1000, then 1000 samples were generated. For MALA we used PyRo's implementation of HMC with stepsize and trajectory length set to 1, as Langevin dynamics is a special case of HMC. We further set the targeted acceptance rate to 0.7 as this is optimal for MALA in 1D (Roberts & Rosenthal, 2001). The samples are compared in Figure 1.

Among the methods, a-GPS demonstrates superior time efficiency (Table 1), requiring only 0.4 seconds compared to 4.6 for NUTS and 2.3 for MALA. This represents a substantial speedup for a-GPS. The Effective Sample Size (ESS) metric, accounting for sample correlation, indicates the quality of samples. Histograms of the effective sample sizes (ESS) are shown in Figure 2: a-GPS significantly outperforms NUTS, yielding an ESS of 954 compared to 364 for NUTS. MALA lags behind with an ESS of 74.9

In order to test our sampler on something where it is never proportional to a mode, we defined a mixture model as $f = \text{lst}(3, 0.2, 1) + \text{Gumbel}(1, 0.6) + \text{Gumbel}(5, 2) + \text{lst}(6, 1, 1)$, where lst is the location-scale t-distribution. We use a-GPS with the initial values $\lambda = 3$, $\mu_0 = 0$, $\sigma_0 = 1$ and $\theta_1 \sim \mathcal{N}(\mu_0, \sigma_0^2)$. The setup for NUTS and MALA is the same as the unimodal example. The samples from one run are shown in Figure 2 (Right). In the mixture scenario, a-GPS remains highly time-efficient, requiring only 2.0 seconds per 1000 samples. This is significantly faster than

Table 1: We report the average time (in seconds) for (burn-in + 1000 samples) and Effective Sample Size (ESS) for 1000 samples over 100 replications. All runs were conducted on the same computer configuration.

| Method | Unimodal (time) ↓ | Unimodal ESS ↑ | Mixture (time) ↓ | Mixture ESS ↑ |
|---|---|---|---|---|
| NUTS (HMC) | 4.6 | 364 | 30.0 | **105.2** |
| MALA (Langevin) | 2.3 | 74.9 | 4.9 | 36.8 |
| a-GPS | **0.4** | **954** | **2.0** | 70.2 |

NUTS (30.0) and MALA (4.9). ESS assumes unimodality, thus its usefulness may be limited in the multimodal setting. However, a-GPS maintains strong sampling quality even in the mixture setting, achieving an ESS of 70.2. While NUTS exhibits a higher ESS (105.2), a-GPS remains competitive and offers notable time savings.

### 3.2 DEEP LEARNING EXPERIMENTS

All methods utilize a single pretrained network as a warm start; refer to Section G for details on obtaining pretrained models. In the main text, tables are vertically split into two sections: the top section comprises methods involving a single forward pass, while the bottom section includes methods that perform a Bayesian model average with multiple forward passes during inference. The inference time (Inference ↓) is highlighted in bold for methods with multiple forward passes.

We adopt the setup from Daxberger et al. (2021) for the Laplace approximation (LA) and refer to Maddox et al. (2019) for the configurations of MC-dropout, SWA, and SWAG. In the case of a-GPS, we maintain the hyperparameters (i.e. batch size, weight decay) as in Maddox et al. (2019), with the exception of replacing the stochastic gradient descent optimizer with our a-GPS method. We set our hyperparameters to $\mu_0 = \theta^{MAP}, \sigma_0 = \lambda$ and the variance limit $\lambda$, we experimented with various values, namely $\lambda = \{1e-4, 1e-5, 1e-6, 1e-7\}$ (refer to Appendix K for a comprehensive table of results). The variance limit remains constant throughout the post-hoc training. Refer to Appendices E and F for details on preventing divergence and the significance of a maximum variance limit. All methods collected the same number of samples, totaling 20 samples for CIFAR10/100, CIFAR5/5, CIFAR50/50, and 45 samples for ImageNet. The experiments employed a batch size of 256 and utilized cross-entropy loss. The evaluation metrics, including Accuracy (Acc), Negative Log Likelihood (NLL), and Expected Calibration Error (ECE), are explained in detail in Appendix H.3.1. These metrics were calculated using the results from a holdout test dataset. The reported values represent the mean ± 1 standard deviation (STD) obtained from five independent runs using pretrained models. For information on the pretrained models, please refer to Section G.

#### 3.2.1 CIFAR10 AND CIFAR100 RESULTS

For CIFAR10, we present the results in Table 2 using the VGG16 architecture (Simonyan & Zisserman, 2015). All methods achieve similar accuracy. We report a-GPS with $\lambda \in \{1e-4, 1e-7\}$. Among the methods with multiple forward-passes at inference MC-Dropout is fastest, likely due to the rapidness of drawing a new sample every forward-pass, unlike SGD-MC and a-GPS that fetch stored models in memory. However, MC-Dropout is limited to dropout architectures. We report LA to be faster than MAP and hypothesize that this is due to the optimisation of LA happening just before inference, thus all or some of the data may be in memory. To determine whether our sampler explored multiple modes, we applied the SWAG model to the samples of a-GPS (denoted as $\lambda$-SWA(G)). Since SWAG constructs a Gaussian approximation of the collected samples, it should yield similar results if the samples are all from one mode. We observe that for a-GPS-7 and 7-SWAG, the results are identical, suggesting that a-GPS-7 stayed in one mode. However, a-GPS-4 and 4-SWAG exhibit a noticeable discrepancy in accuracy and a significant difference in ECE and NLL (Table 2). This suggests that a-GPS-4 has traversed different modes separated by relatively high loss (low accuracy) areas.

In Table 3, we present the results for CIFAR100 obtained using the WideResNet28-10 architecture (Zagoruyko & Komodakis, 2017). Once again, we observe indications that a-GPS-4 explored more than just a single mode, as evidenced by the performance drop in 4-SWA(G). This exploration

Table 2: Performance results for VGG16 on CIFAR10. Mean ± 1STD of 5 runs are reported. MAP represents the performance of five pretrained models. MC-drop is the dropout method, and Laplace (LA). a-GPS-$\lambda$-SWA(G) is abbreviated as $\lambda$-SWA(G). $\lambda$-SWA(G) is the result of building a SWA(G) model with a-GPS samples. SGD-MC is stochastic gradient descent used as a sampler. Inference ↓ denotes the relative inference time to MAP.

| Method | Acc ↑ | ECE ↓ | NLL ↓ | Inference ↓ |
|---|---|---|---|---|
| MAP | 93.02 ± 0.19 | 4.88 ± 0.26 | 0.34 ± 0.01 | 1.0 |
| LA | 93.04 ± 0.16 | 2.66 ± 0.23 | 0.25 ± 0.01 | 0.16 |
| SWA | 93.18 ± 0.16 | 4.18 ± 0.15 | 0.27 ± 0.01 | 1.09 |
| 7-SWA | 93.06 ± 0.17 | 4.96 ± 0.22 | 0.35 ± 0.01 | 1.09 |
| 4-SWA | 39.58 ± 18.62 | 60.3 ± 18.7 | 9.63 ± 3.01 | 1.09 |
| | | | | |
| MC-drop | 93.02 ± 0.19 | 4.44 ± 0.26 | 0.31 ± 0.01 | **2.06** |
| SGD-MC | **93.22** ± 0.23 | 1.30 ± 0.24 | **0.21** ± 0.00 | 2.85 |
| SWAG | 93.20 ± 0.18 | **1.08** ± 0.24 | **0.21** ± 0.00 | 10.91 |
| 7-SWAG | 93.04 ± 0.21 | 4.96 ± 0.22 | 0.35 ± 0.01 | 10.91 |
| 4-SWAG | 45.06 ± 18.63 | 25.0 ± 5.72 | 2.08 ± 0.76 | 10.91 |
| a-GPS-4 | 91.64 ± 0.12 | 43.6 ± 4.96 | 0.85 ± 0.10 | 2.85 |
| a-GPS-7 | 93.04 ± 0.21 | 4.96 ± 0.22 | 0.35 ± 0.01 | 2.85 |

Table 3: CIFAR100 results with WideResNet28-10. a-GPS-4-SWA(G) is denoted as 4-SWA(G). SGD-MC is used as a sampler for SWA(G).

| Method | Acc ↑ | ECE ↓ | NLL ↓ | Inference ↓ |
|---|---|---|---|---|
| MAP | 79.50 ± 0.30 | 5.94 ± 0.27 | 0.89 ±0.01 | 1.0 |
| LA | 79.40 ± 0.35 | 18.52 ± 0.65 | 1.02 ± 0.02 | 17.93 |
| SWA | **80.64** ± 0.27 | 6.76 ± 0.08 | 0.77 ± 0.01 | 1.12 |
| 4-SWA | 77.70 ± 0.28 | 9.56 ± 0.70 | 0.97 ± 0.02 | 1.12 |
| | | | | |
| SGD-MC | 80.44 ± 0.24 | 9.34 ± 0.25 | 0.76 ± 0.01 | **2.89** |
| SWAG | 80.14 ± 0.21 | 4.80 ± 0.60 | **0.73** ± 0.01 | 10.80 |
| 4-SWAG | 74.92 ± 0.15 | 46.76 ± 1.48 | 1.72 ± 0.05 | 10.80 |
| a-GPS-4 | 78.64 ± 0.26 | **2.60** ± 0.33 | 0.78 ± 0.01 | **2.89** |
| a-GPS-5 | 79.82 ± 0.31 | 3.92 ± 0.28 | 0.82 ± 0.01 | **2.89** |
| a-GPS-7 | 79.86 ± 0.36 | 5.40 ± 0.19 | 0.86 ± 0.01 | **2.89** |

appears to have benefited a-GPS-4, as its Expected Calibration Error (ECE) is significantly lower than that of the other methods.

### 3.2.2 IMAGENET RESULTS

For the ImageNet2012 classification dataset (Deng et al., 2009), we used PyTorch's (Paszke et al., 2019) recipe for ResNet-50-V1[1] (He et al., 2016). This training procedure was replicated five times with five different seeds to obtain 5 different pretrained models. We followed Maddox et al. (2019)'s procedure for obtaining the SWA(G) model on ImageNet. For a-GPS, we chose the variance limits $\lambda = \{1e-4, 1e-5, 1e-6\}$. Since the CIFAR10/100 experiments suggested that a-GPS explored the landscape, we also ran experiments with a-GPS for only one epoch, effectively decreasing training time 10x (Train ↓ denotes the post-hoc training epochs).

In Table 4, LA suffers compared to MAP, especially with regards to ECE, suggesting that the mode of MAP is non-Gaussian. SWAG also constructs a Gaussian, though it may have a smaller variance than LA. The best is SGD-MC; the good performance may come from traversing around the top of a mode, which may not be Gaussian. A-GPS performs well, even for a relatively large variance limit. We do emphasize that a-GPS does not optimize a loss to achieve good performance, it draws a random sample from the approximation of the landscape. Notice, however, that a-GPS-1-4 does better than a-GPS-4; perhaps a-GPS-4 diverged slightly because of too much noise over time. In

---

[1]github.com/pytorch/vision/tree/main/references/classification

Table 4: Metrics on ImageNet with the ResNet-50 architecture. The 1-$\lambda$ denotes the results for when a-GPS was only run for a single epoch instead of 10.

| Method | Acc ↑ | ECE ↓ | NLL ↓ | Inference ↓ | Train ↓ |
|---|---|---|---|---|---|
| MAP | 76.10 ± 0.06 | 3.34 ± 0.08 | 0.95 ± 0.004 | 1 | - |
| LA | 75.86 ± 0.10 | 15.32 ± 0.12 | 1.05 ± 0.004 | 28.69 | 1 |
| SWA | 76.45 ± 0.05 | 2.15 ± 0.05 | 0.93 ± 0.002 | 12.36 | 10 |
| 1-4-SWA | 66.56 ± 0.85 | 6.86 ± 0.26 | 1.62 ± 0.053 | 12.36 | 1 |
| | | | | | |
| SGD-MC | **76.58** ± 0.13 | **1.88** ± 0.04 | **0.91** ± 0.004 | **14.31** | 10 |
| SWAG | 76.50 ± 0.10 | 5.05 ± 0.15 | 0.94 ± 0.001 | 370.01 | 10 |
| a-GPS-4 | 70.74 ± 0.12 | 8.14 ± 0.15 | 1.25 ± 0.004 | **14.31** | 10 |
| a-GPS-1-4 | 73.98 ± 0.10 | 5.04 ± 0.05 | 1.08 ± 0.006 | **14.31** | **1** |
| a-GPS-6 | 76.14 ± 0.08 | 3.70 ± 0.09 | 0.95 ± 0.004 | **14.31** | 10 |
| a-GPS-1-6 | 76.12 ± 0.07 | 3.60 ± 0.00 | 0.95 ± 0.004 | **14.31** | **1** |

Table 5: Predictive entropies for the CIFAR5-5 and WideResNet28-10, in (IND) and out (OOD) of distribution. For IND, lower is better; for OOD, higher is better.

| Method | IND ENT ↓ | OOD ENT ↑ |
|---|---|---|
| MAP | 0.08 ± 0.003 | 0.67 ± 0.016 |
| LA | 0.12 ± 0.004 | 0.67 ± 0.016 |
| SWA | 0.08 ± 0.003 | 0.54 ± 0.008 |
| | | |
| SGD-MC | 0.13 ± 0.059 | **0.85** ± 0.151 |
| SWAG | 0.12 ± 0.039 | 0.77 ± 0.186 |
| a-GPS-4 | 0.08 ± 0.005 | 0.60 ± 0.016 |
| a-GPS-5 | **0.07** ± 0.003 | 0.64 ± 0.012 |
| a-GPS-6 | **0.07** ± 0.003 | 0.66 ± 0.012 |

Table 6: Predictive entropies for the CIFAR50-50 and WideResNet28-10, in (IND) and out (OOD) of distribution. For IND, lower is better; for OOD, higher is better.

| Method | IND ENT ↓ | OOD ENT ↑ |
|---|---|---|
| MAP | 0.98 ± 0.105 | 1.08 ± 0.011 |
| LA | 2.99 ± 0.060 | 1.91 ± 0.015 |
| SWA | 0.67 ± 0.045 | **3.42** ± 0.318 |
| | | |
| SGD-MC | 1.38 ± 0.142 | 1.04 ± 0.022 |
| SWAG | 1.28 ± 0.171 | 3.17 ± 0.112 |
| a-GPS-4 | **0.64** ± 0.058 | 2.28 ± 0.119 |
| a-GPS-5 | 0.85 ± 0.060 | 2.38 ± 0.127 |
| a-GPS-6 | 1.04 ± 0.102 | 2.95 ± 0.189 |

addition, a-GPS-1-4 may have explored different modes in a short amount of time since 1-4-SWA(G) gets a significant decrease in performance. All results are reported in Appendix K.

The average inference time (Inference ↓) of the five runs shows that the inference time for big datasets is an issue for SWA(G). The time of SWA is comparable to a-GPS, even though SWA only does one forward pass, and a-GPS does 45. While SWAG only does 30 forward passes, a-GPS is still 25-30x faster. For more information about why SWA(G) has such a slow inference time, see Appendix H.

## 3.3 OUT-OF-DISTRIBUTION (OOD)

In order to construct an OOD test for the CIFAR10/100 datasets, the models were trained with the exact same hyperparameters as in Section 3.2.1. However, the data was split class-wise into two equally sized groups; (0, 1, 2, 8, 9) and (3, 4, 5, 6, 7) for CIFAR10 following Maddox et al. (2019), and $0 \rightarrow 49$ and $50 \rightarrow 99$ for CIFAR100. This gives us two new datasets CIFAR5-5 and CIFAR50-50. The models were *only trained on half of the classes*. We report the predictive entropy, defined as

$$\text{ENT} = -\sum_D f(D_x) \log f(D_x). \tag{13}$$

The validation set is also split in the same manner. To test the predictive entropy for in-distribution data (IND ENT), we use the partition of the validation dataset with the same classes that were used during training. For the predictive entropy on OOD (OOD ENT), we use the partition of classes from the validation dataset that was *not* used during training. All experiments were replicated 5 times to provide the mean and standard deviation of the total predictive entropy for IND and OOD. We report results for the WideResNet28-10 see Appendix K for more results.

In Table 5, a-GPS-5 and a-GPS-6 are both certain about IND and are relatively uncertain about OOD; this is desired behavior. The other methods also exhibit similar behavior, though SWA has a smaller relative difference. SGD-MC is most uncertain for OOD.

From Table 6, we observe undesirable behavior from SGD-MC and LA as they exhibit higher uncertainty for IND than OOD. SGD-MC relies on the inherent noise present in the data, and as a result, fluctuations in the training set may lead to inconsistent uncertainty estimates. When the assumptions of LA do not align with the true distribution, especially in complex and multimodal scenarios, LA fails to accurately capture the underlying uncertainty. In the case of our experiments, both SGD-MC and LA's inconsistency in uncertainty estimates for in-distribution and out-of-distribution data suggests a failure to adapt to the true data distribution. Once again, a-GPS demonstrates greater certainty about IND and relatively higher uncertainty for OOD, though SWA exhibits the highest uncertainty for OOD.

## 4 CONCLUSION

In conclusion, our study highlights the Adaptive Proposal Sampling framework, particularly through the approximate Gaussian proposal sampler (a-GPS). A-GPS excels in unimodal scenarios, demonstrating fast parameter updates and high-quality sampling. Even in the face of challenging mixture distributions, it maintains a good balance between speed and effectiveness, positioning itself as a versatile tool for Bayesian inference tasks.

A-GPS exhibits computational efficiency during training, rivaling SGD in speed, making it a feasible option for time-sensitive tasks. Compared to established methods like HMC and Langevin dynamics, a-GPS showcases superior computational speed while maintaining a substantial effective sample size, especially within modes. In deep learning contexts, a-GPS achieves comparable results to SWA, SWAG, and SGD-MC with significantly reduced training time, suggesting notable efficiency gains. During inference, a-GPS demonstrates linear computational complexity with the number of samples. Effectively scaling to large datasets, unlike SWA and SWAG, it maintains a balanced trade-off between computational speed and model exploration.

The observed behaviors in methods like SGD-MC and LA underscore the need for a nuanced and adaptive approach to uncertainty modeling. Unlike a-GPS, these conventional methods may lack the flexibility required to navigate diverse data distributions and could oversimplify complex structures inherent in real-world datasets. A-GPS, with its adaptability and exploration capabilities, emerges as a promising alternative that aligns well with the intricacies of uncertainty in various scenarios.

While not surpassing standards universally, the model showcases competitiveness as a novel sampler, swiftly generating a posterior approximation for any model. Its multimodal capability, coupled with a high effective sample size within modes, underscores the need for further exploration and elaboration.

In summary, a-GPS emerges not just as a solution to uncertainty modeling challenges but as a catalyst for advancing the broader field of machine learning. Its adaptability, efficiency, and potential for exploration position it as a promising alternative in the pursuit for robust Bayesian inference methodologies.

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

## A  BAYESIAN SETTING AND DEEP LEARNING

Bayesian methods aim to find the posterior distribution of the parameters vector $\theta$:

$$p(\theta|D) = \frac{p(D|\theta)p(\theta)}{p(D)}, \tag{14}$$

where $p(\theta)$ is the prior, $D$ represents the data, $p(D|\theta)$ is the likelihood, and $p(D)$ is the normalizing factor. Using the posterior, we can calculate the posterior predictive distribution for a new data point $x^*$:

$$p(y^*|x^*, D) = \int p(y^*|x^*, \theta)p(\theta|D)d\theta. \tag{15}$$

While this integral is often intractable, it can be approximated with a Monte Carlo estimate:

$$p(y^*|x^*, D) \approx \sum_S p(y^*|x^*, \theta_s), \tag{16}$$

where $\theta_s \sim p(\theta|D)$ is a sample from the posterior. Various Bayesian methods have their own approaches to obtaining the posterior $p(\theta|D)$.

Score matching (Hyvärinen, 2005) utilizes the fact that the posterior $p(\theta|D)$ has a log gradient with respect to the input $\theta$ that is independent of the normalizing factor $p(D)$. The $\nabla \log p(\theta|D)$ is referred to as the score function. Hyvärinen (2005) proposes optimizing a score matching objective to find the parameters of interest. We suggest parameter updates such that a chosen proposal distribution $q(\theta)$ converges to match the score function of the posterior.

In the context of deep learning, the objective is to minimize a loss function with respect to the parameters given the data, $\mathcal{L}_{D_x, D_y}(\theta)$, where $D_x$ and $D_y$ represent the data-label pairs of the dataset. In the Bayesian perspective, the loss function is interpreted as the negative log likelihood, and $L_2$-regularization ($||\theta||$) serves as a Gaussian prior. Thus, the posterior can be expressed as:

$$f(\theta \mid D_x, D_y) \propto \exp(-\mathcal{L}_{D_x, D_y}(\theta)) \exp(-||\theta||). \tag{17}$$

This leads to:

$$\log f(\theta \mid D_x, D_y) = -(\mathcal{L}_{D_x, D_y}(\theta) + ||\theta||). \tag{18}$$

## B  CONVERGENCE OF GAUSSIAN PROPOSAL

We can expand the update for $\mu_{t+1}$ as

$$\mu_{t+1} \mid \sigma_{t+1}^2 = \theta_t + \sigma_{t+1}^2 \times \nabla \log f(\theta_t) \tag{19a}$$

$$= \theta_t + \left| \frac{\theta_t - \mu_t}{\nabla \log f(\theta_t)} \right| \times \nabla \log f(\theta_t) \tag{19b}$$

$$= \theta_t + \text{sgn}\left[\nabla f\right] \times |\theta_t - \mu_t|, \tag{19c}$$

where $\text{sgn}[\nabla f]$ is the sign function of $\nabla \log f(\theta_t)$, and $|\cdot|$ is the absolute value.

In the case where $f \propto \mathcal{N}(\mu, \sigma^2)$ (i.e. $g(\theta) = \frac{-(\theta - \mu)^2}{2\sigma^2}$), we have that

$$\sigma_{t+1}^2 = \sigma^2 \left| \frac{\theta_t - \mu_t}{\theta_t - \mu} \right|, \tag{20}$$

thus the variance estimate is unbiased whenever $\mu_t = \mu$. However, when $\mu_t \neq \mu$ the true variance is likely to be underestimated as $\theta_t \sim q = \mathcal{N}(\mu_t, \sigma_t^2)$ likely is closer to the mean of the distribution it was drawn from. It then follows that we are likely to underestimate the true variance

$$\sigma_{t+1}^2 = \sigma^2 \left| \frac{\theta_t - \mu_t}{\theta_t - \mu} \right| < \sigma^2. \tag{21}$$

When it comes to the update of $\mu_{t+1}$, we have that

$$\text{sgn}[\nabla f] = \text{sgn}\left[-\frac{\theta_t - \mu}{\sigma^2}\right] = \text{sgn}[-\theta_t + \mu], \tag{22}$$

since we assume $\sigma^2 > 0$. Thus the sign of the gradient $\text{sgn}[\nabla f]$ indicates to which side the true $\mu$ lies in relation to our current sample $\theta_t$,

$$\text{sgn}[\nabla f] = \begin{cases} 1 & \theta_t < \mu \\ -1 & \theta_t > \mu \\ 0 & \theta_t = \mu. \end{cases} \tag{23}$$

If $\text{sgn}[\nabla f]$ is the same as $\text{sgn}[-(\theta_t - \mu_t)] = \text{sgn}[\nabla q_{t-1}(\theta_t)]$ (onward denoted as $\text{sgn}[\nabla q]$) then we do not change our estimate of $\mu$. For example if $\text{sgn}[\nabla q] = \text{sgn}[\nabla f]$ then

$$\mu_{t+1} = \theta_t + \text{sgn}[\nabla q]|\theta_t - \mu_t| = \mu_t. \tag{24}$$

On the other hand if the direction of the true mean is different from our estimate, $\text{sgn}[\nabla f] = -\text{sgn}[\nabla q]$, our estimate will then change to

$$\mu_{t+1} = \theta_t - sqn[\nabla q]|\theta_t + \mu_t| \neq \mu_t. \tag{25}$$

The case when $\text{sgn}[\nabla f] = -\text{sgn}[\nabla q] \neq 0$ only happens when we have sampled a $\theta_t$ that lies between the true $\mu$ and the current estimate $\mu_t$:

$$\theta_t \in \begin{cases} (\mu_t, \mu) & \theta_t > \mu_t \\ (\mu, \mu_t) & \theta_t < \mu_t. \end{cases} \tag{26}$$

It follows that

$$0 < \frac{\theta_t - \mu}{\mu_t - \mu} < 1. \tag{27}$$

Where the probability to get as close or closer to $\mu$ is (in the case $\mu_t < \mu$)

$$P(|\mu_{t+1} - \mu| \leq |\mu_t - \mu|) = \int_{\mu_t}^{\mu} q(\theta)d\theta. \tag{28}$$

In the edge cases when $\theta_t = \mu$ then Equation (19) gives $\mu_{t+1} = \mu$ because $\text{sgn}[\nabla f] = 0$, and also if $\theta_t = \mu_t$ then Equation (19) gives $\mu_{t+1} = \mu_t$ because $|\theta_t - \mu_t| = 0$. Thus, $\mu_{t+1}$ will always be as close or closer to $\mu$.

This gives us convergence of both parameters as the estimate of $\mu_{t+1} \xrightarrow{t \to \infty} \mu$, from Equation (7) we also have a convergence of the variance

$$\sigma_{t+1}^2 = \left| (\theta_t - \mu_t) \times \frac{1}{\nabla \log f(\theta_t)} \right|, \tag{29a}$$

$$\sigma_{t \to \infty}^2 = \left| (\theta_t - \mu) \times \frac{\sigma^2}{-(\theta_t - \mu)} n \right| \tag{29b}$$

$$= \sigma^2. \tag{29c}$$

Thus the Gaussian proposal $q$ converges to match the score function to any $f \propto \mathcal{N}(\mu, \sigma)$.

## C   EXACT GAUSSIAN PROPOSAL SAMPLER

The exact GPS uses the MH correction and it takes into consideration that the proposals are not symmetric i.e. $q(\theta * |\theta_t) \neq q(\theta_t|\theta*)$. Proof of convergence can be found in Appendix D.

## D   CONVERGENCE

To ensure that the stationary distribution of the Markov chain is the target distribution, it needs to be reversible with respect to the target (Metropolis et al., 1953; Hastings, 1970; Rosenthal, 2011): $f(i)p(j|i) = f(j)p(i|j)$. The probability of going from state $i$ to state $j$ can be deconstructed

---

**Algorithm 2** Exact Gaussian Proposal Sampler

---

**Input:** Variance limit= $\lambda$, $\mu_0 = 0$, $\sigma_0 = \lambda$
$\theta_1 \sim \mathcal{N}(\mu_0, \sigma_0^2)$

**for** $t = 1$ **to** $T$ samples **do**
$\sigma_{t+1}^2 \leftarrow \left| \frac{(\theta_t - \mu_t)}{\nabla \log f(\theta_t)} \right|$
$\mu_{t+1} \leftarrow \theta_t + \sigma_{t+1}^2 \times \nabla \log f(\theta_t)$

$\sqrt{\sigma_{t+1}^2} = \min(\sqrt{\sigma_{t+1}^2}, \lambda)$
$\theta^* \sim q(\cdot \mid \theta_t) = \mathcal{N}(\mu_{t+1}, \sigma_{t+1}^2)$

$\sigma^{2*} \leftarrow \left| \frac{(\theta^* - \mu_{t+1})}{\nabla \log f(\theta^*)} \right|$
$\mu^* \leftarrow \theta^* + \sigma^{2*} \times \nabla \log f(\theta^*)$

$\sqrt{\sigma^{2*}} = \min(\sqrt{\sigma^{2*}}, \lambda)$
$q(\cdot \mid \theta^*) = \mathcal{N}(\mu^*, \sigma^{2*})$

Metropolis-Hastings correction:
Accept $\theta_{t+1} = \theta^*$ with probability $\alpha(\theta_t, \theta^*) = \min\{1, \frac{f(\theta^*)q(\theta_t|\theta^*)}{f(\theta_t)q(\theta^*|\theta_t)}\}$ otherwise $\theta_{t+1} = \theta_t$
**end for**

---

$p(j|i) = q(j|i)\alpha(i,j)$, where $q(j|i)$ is probability of sampling $j$ from state $i$, and $\alpha(j,i)$ is the probability of accepting the proposed state $j$.

In our case it can happen that $q(j|i) \neq q(i|j)$, so the general Metropolis-Hastings acceptance rule (Hastings, 1970) which can be written as

$$\alpha(i,j) = \min\{1, \frac{f(j)q(i|j)}{f(i)q(j|i)}\}, \tag{30}$$

has to be applied in order to guarantee reversibility,

Given this acceptance rule it follows that if $\alpha(i,j) < 1$, then $\alpha(j,i) \geq 1$ so

$$f(i)p(j|i) = f(j)p(i|j) \tag{31a}$$
$$f(i)q(j|i)\alpha(i,j) = f(j)q(i|j)\alpha(j,i) \tag{31b}$$
$$f(i)q(j|i)\frac{f(j)q(i|j)}{f(i)q(j|i)} = f(j)q(i|j). \tag{31c}$$

Conversely, if $\alpha(i,j) \geq 1$, then $\alpha(j,i) < 1$ so

$$f(i)p(j|i) = f(j)p(i|j) \tag{32a}$$
$$f(i)q(j|i)\alpha(i,j) = f(j)q(i|j)\alpha(j,i) \tag{32b}$$
$$f(i)q(j|i) = f(j)q(i|j)\frac{f(i)q(j|i)}{f(j)q(i|j)}. \tag{32c}$$

## E  HITTING MINIMA IN THE SEARCH FOR MAXIMA

Assuming the Gaussian form, there exist no minimum. However, the true target distribution may consist of multiple modes, thus it has at least one minimum. Whenever, $\theta_t$ is close to a minimum, the proposal distribution mimics a Gaussian that has its peak close to the minimum. This resulting Gaussian may have a very high variance, and in order to limit this pitfall the authors suggest to set a maximum limit $\lambda$ on the proposed variance, $\sqrt{\sigma_{t+1}^2} = \min\left(\sqrt{\sigma_{t+1}^2}, \lambda\right)$. The choice of a variance

limit $\lambda$ may be specific to each problem, and this will of course limit the exploration if the true distribution has a high variance. However, for deep learning it may be beneficial to have a more conservative limit.

## F    CHOOSING THE VARIANCE LIMIT FOR DEEP LEARNING

In deep learning where there are millions of parameters, the amount of noise induced can be too much and the sampler may diverge. In order to find a variance limit, we tried different limits and observed the training loss for some batches. If the training loss diverged then the variance limit was too high. In the other end if it is too low then it might not explore. In the end we chose a set of variance limits $\lambda = \{1e-4, 1e-5, 1e-6, 1e-7\}$ that did not diverge in training loss. For some limits the training loss and accuracy increased compared to the pretrained starting point, other times it decreased, but not too drastically.

## G    PRETRAINED MODELS CIFAR10 AND CIFAR100

In order to obtain pretrained models for both CIFAR10 and CIFAR100 (Krizhevsky, 2009) we follow Maddox et al. (2019)'s setup for for training VGG16 (Simonyan & Zisserman, 2015) and WideResNet28-10 (Zagoruyko & Komodakis, 2017). WideResNet28-10 was trained for 300 epochs with stochastic gradient descent and an initial learning rate of 0.1, weight decay at 0.0005, batch size 128 and momentum 0.9. Cross-entropy was used as the loss function. For VGG16 we followed the same procedure, except a lower initial learning rate of 0.05, because of the absence of batch-normalization. After 150 epochs, a learning rate schedule started so that it would decrease towards 0.0005 at epoch 270, then finish the last 30 epochs with the learning rate set to 0.0005, see (Maddox et al., 2019; Izmailov et al., 2019) for more details. We also included ResNet-50(He et al., 2016) and followed the same training setup as WideResNet28-10. These training runs were repeated five times with five different seeds in order to get five different pretrained models, which will work as the starting point for our post-hoc methods.

## H    POST-HOC METHOD SETUP

### H.1    LAPLACE ESTIMATE

Since the Laplace approximation requires a MAP estimate we gave it our pretrained model (Appendix G). For the results we used Daxberger et al. (2021)'s Laplace PyTorch library, and followed their recommendations. The Laplace approximation was applied to the last-layer in the pretrained network, with KFAC estimation (Martens & Grosse, 2015) of the Hessian matrix. The Hessian estimation was based on the whole training dataset. Then the prior precision was optimized in accordance to their recommendation, see (Daxberger et al., 2021) for more information. Due to the $O(n^2)$ memory requirement of LA we experienced out-of-memory errors for the VGG-16 architecture on CIFAR100, we had to deviate from the standard recommendations, by changing prediction type to 'nn' and link approximation to 'mc'

### H.2    MC-DROPOUT

To get the MC-dropout estimate, we have to enable dropout at test-time (Gal & Ghahramani, 2016) and computed the same number of forward passes as a-GPS (20 for CIFAR10/100 and 45 for ImageNet). These forward passes were averaged to obtain the Bayesian model average that was used at test time. The MC-dropout results are only presented together with VGG16, as it is the only architecture that we tested that also has dropout-layers.

### H.3    SWA AND SWAG

For SWA and SWAG we follow the procedure of (Maddox et al., 2019), for a pretrained network. To collect the models for the CIFAR10/100 experiments, we used the MAP as a starting point. We continue running the stochastic gradient descent for another 20 epochs, but changed the learning

rate to 0.02 as this was the default in their python library. The final SWA(G) model consists of a the models collected at the end of every epoch, in addition to the initialization with the MAP model. Finally, for the SWAG model a low rank covariance matrix was estimated using their suggested rank=20. We also follow Maddox et al. (2019) for the ImageNet experiments: 11 epochs, collect 4 models per epoch. The SGD optimizer setup was momentum 0.9, learning rate 0.001 and weight decay at 0.0005. For the SWA predictions the 20 models (CIFAR10/100) collected were averaged to become the SWA model. Then a single forward pass produced the predictions. For the SWAG prediction on CIFAR10/100 we had 20 forward passes.

For ImageNet we follow Maddox et al. (2019)'s procedure for obtaining the SWA(G) model using stochastic gradient descent with a constant learning rate of $1e-3$, rank K = 20, weight decay at $5e-4$, momentum 0.9, and batch size 256. Using our pretrained ResNet-50 model as the initialization, additional models were collected 4 times per epoch. We did this for 11 epochs to collect a total of 45 models.

For best prediction accuracy Maddox et al. (2019) update the batch norm of ResNet-50 with a sample of 10% of the training dataset for the SWA model. Maddox et al. (2019); Izmailov et al. (2019) point out the SWA(G) model is only collected during the post-training and the batch norm parameters need to be updated when this model is going to be used for inference, because the model has not directly seen the data.

For SWAG Maddox et al. (2019) suggest to continually update the batch norm parameters with 10% of the training data for each sample. Because of the slow inference time we limited SWAG to draw 30 samples from the SWAG distribution. It is also worth mentioning that for SWA(G) we followed the batch-normalization boosting schedule that Maddox et al. (2019); Izmailov et al. (2019) found to be very effective. By updating the batch-normalization parameters after sampling a new set of weights with 10% of the training data (30 samples corresponds to 300% of training data, or 3 epochs). Maddox et al. (2019) show in their appendix D.4 that the negative log likelihood is halved by this batch-norm boosting.

### H.3.1 THE METRICS USED

Standard metrics reported in the results include the mean ± 1 standard deviation (STD) of five runs from independent pretrained models (for details on our pretrained models, refer to Appendix G). The metrics were computed using the holdout test dataset denoted as $D^*$, where $(D_x^*, D_y^*)$ represents a data-label pair.

Accuracy (Acc): This metric measures the percentage of correct predictions:

$$\text{Acc} = \frac{100}{|D^*|} \sum_{D^*} \mathbf{1}_{f(D_x^*)=D_y^*}. \tag{33}$$

Negative Log Likelihood (NLL): Calculated as:

$$\text{NLL} = -\sum_{D^*} \log f_{D_y^*}(D_x^*). \tag{34}$$

Expected Calibration Error (ECE): This metric involves partitioning the $M$ classes into bins $(B_m)$ based on accuracy and predicted softmax score, called confidence. It quantifies the difference between the model's accuracy for a class and its confidence (Conf):

$$\text{ECE} = -\sum_{m=1}^{M} \frac{|B_m|}{n} |\text{Acc}(B_m) - \text{Conf}(B_m)|. \tag{35}$$

## I  OTHER ADAPTIVE PROPOSAL SAMPLERS

This section provides parameter updates for different choices of proposal: Gamma and Beta. The theoretical foundation for these methods are not as thorough as for the Gaussian Proposal Sampler. Future research, may find a set of updates that provide convergence guarantees and a stability parameter like the Gaussian variance limit. However, we propose these APS updates for the Gamma and Beta, in order to show that the APS framework is applicable to several cases.

## I.1 GAMMA PROPOSAL SAMPLER

Set the proposal distribution $q = \text{Gamma}$, and assume that the modes in the target distribution also is Gamma. We have that

$$\nabla \log f(\theta_t) = \nabla \log q(\theta_t) \tag{36a}$$

$$= \frac{\alpha_{t+1} - 1}{\theta} - \beta_{t+1}. \tag{36b}$$

Using this information we propose the following updates:

$$\alpha_{t+1} = |(\nabla \log f(\theta_t) + \beta_t) \cdot \theta_t + 1| \tag{37a}$$

$$\beta_{t+1} = |\frac{\alpha_t - 1}{\theta_t} - \nabla \log f(\theta_t)|. \tag{37b}$$

In this case $\alpha$ and $\beta$ does not interact, we can simply use this equation to extract the parameters and we get the updates

---

**Algorithm 3** Approximate Gamma Proposal Sampler

---

**Input:** $\alpha_0 = 1$, $\beta_0 = 1$,
$\theta_0 \sim \text{Gamma}(\alpha_0, \beta_0^2)$
**for** $t = 0$ **to** $T$ samples **do**
   $\alpha_{t+1} \leftarrow |(\nabla \log f(\theta_t) + \beta_t) * \theta_t + 1|$
   $\beta_{t+1} \leftarrow |-\nabla \log f(\theta_t) + \frac{(\alpha_t - 1)}{\theta_t}|$
   $\theta^{t+1} \sim \text{Gamma}(\alpha_{t+1}, \beta_{t+1})$
**end for**

---

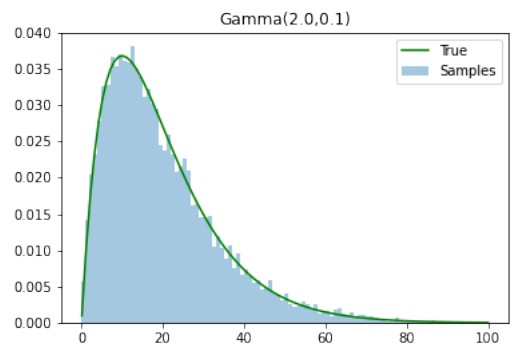

Figure 3: Approximate Gamma Proposal Sampler on when $f = \text{Gamma}(2, 0.1)$.

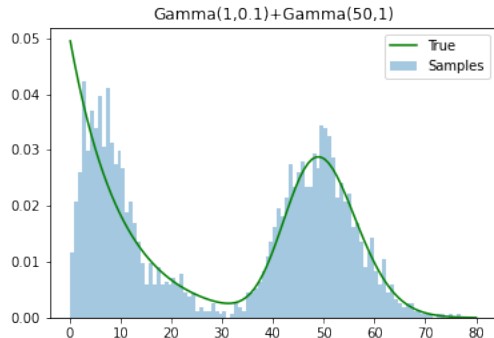
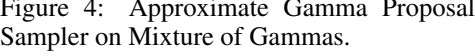

Figure 4: Approximate Gamma Proposal Sampler on Mixture of Gammas.

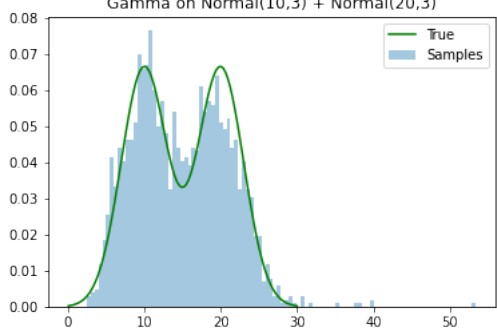

Figure 5: Approximate Gamma Proposal Sampler on mixture of Gaussians

## J  BETA PROPOSAL SAMPLER

Set the proposal distribution $q = \text{Beta}$, and assume that the modes in the target distribution also is Beta. We have that

$$\nabla \log f(\theta) = \nabla \log q(\theta) \tag{38a}$$

$$= \frac{\alpha - 1}{\theta} - \frac{\beta - 1}{1 - \theta}. \tag{38b}$$

Using this information we propose the following updates:

$$\alpha_{t+1} = \left| \left( \nabla \log f(\theta) + \frac{\beta_t - 1}{1 - \theta} \right) \theta_t + 1 \right| \tag{39a}$$

$$\beta_{t+1} = \left| - \left( \nabla \log f(\theta) - \frac{\alpha_t - 1}{\theta} \right) (1 - \theta_t) + 1 \right|. \tag{39b}$$

---

**Algorithm 4** Approximate Beta Proposal Sampler

---

**Input:** $\alpha_0 = 1, \beta_0 = 1$
$\theta_0 = 0.5$
**for** $t = 0$ **to** $T$ samples **do**
  $\alpha_{t+1} \leftarrow |(\nabla \log f(\theta_t) + \frac{(\beta_t - 1)}{1 - \theta_t})\theta + 1|$
  $\beta_{t+1} \leftarrow | - (\nabla \log f(\theta_t) - \frac{\alpha_t - 1}{\theta_t})(1 - \theta) + 1|$
  $\theta^{t+1} \sim \text{Beta}(\alpha_{t+1}, \beta_{t+1})$
**end for**

---

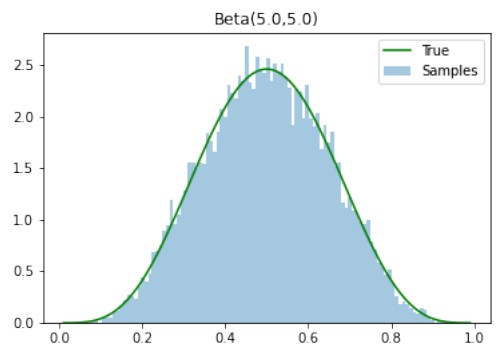

Figure 6: Approximate Beta proposal sampler on a Beta(5,5) distribution.

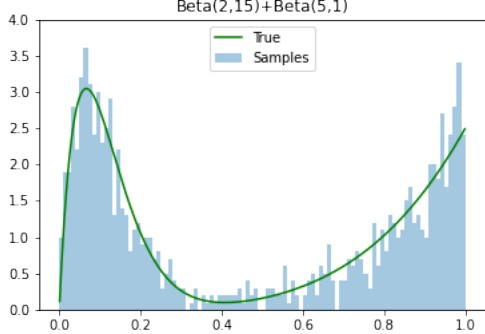

Figure 7: Approximate Beta proposal sampler on a mixture of Betas.

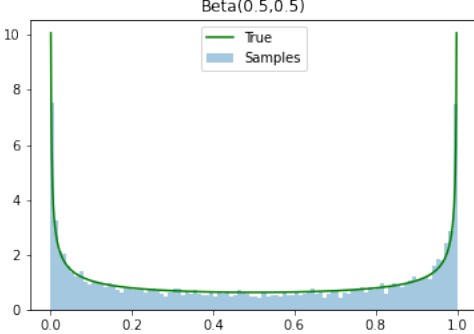

Figure 8: Approximate Beta proposal sampler on a Beta(0.5,0.5) distribution

## K  RESULTS

Table 7: CIFAR10 Metrics for three different architectures.

| Model | Method | Acc ↑ | ECE ↓ | NLL ↓ |
|---|---|---|---|---|
| ResNet-50 | MAP | 90.15 ± 0.21 | 7.00 ± 0.16 | 0.46 ± 0.011 |
| ResNet-50 | LA | 90.15 ± 0.21 | 5.83 ± 0.19 | 0.39 ± 0.008 |
| ResNet-50 | a-GPS-4 | 89.82 ± 0.20 | 6.04 ± 0.22 | 0.48 ± 0.016 |
| ResNet-50 | a-GPS-4-SWA | 89.62 ± 0.34 | 9.04 ± 0.30 | 0.88 ± 0.045 |
| ResNet-50 | a-GPS-4-SWAG | 89.78 ± 0.17 | 3.98 ± 0.30 | 0.38 ± 0.009 |
| ResNet-50 | a-GPS-5 | 90.16 ± 0.35 | 7.62 ± 0.32 | 0.54 ± 0.012 |
| ResNet-50 | a-GPS-5-SWA | 90.06 ± 0.36 | 8.06 ± 0.29 | 0.61 ± 0.015 |
| ResNet-50 | a-GPS-5-SWAG | 90.12 ± 0.34 | 7.60 ± 0.30 | 0.54 ± 0.013 |
| ResNet-50 | a-GPS-6 | 90.12 ± 0.30 | 7.08 ± 0.25 | 0.47 ± 0.010 |
| ResNet-50 | a-GPS-6-SWA | 90.10 ± 0.24 | 7.22 ± 0.23 | 0.48 ± 0.011 |
| ResNet-50 | a-GPS-6-SWAG | 90.12 ± 0.30 | 7.14 ± 0.26 | 0.47 ± 0.010 |
| ResNet-50 | a-GPS-7 | 90.12 ± 0.29 | 6.96 ± 0.26 | 0.46 ± 0.010 |
| ResNet-50 | a-GPS-7-SWA | 90.08 ± 0.23 | 7.08 ± 0.23 | 0.46 ± 0.011 |
| ResNet-50 | a-GPS-7-SWAG | 90.16 ± 0.27 | 6.96 ± 0.25 | 0.46 ± 0.010 |
| ResNet-50 | SGD-MC | 90.74 ± 0.21 | 3.24 ± 0.17 | **0.29** ± 0.003 |
| ResNet-50 | SWA | **90.82** ± 0.19 | 4.22 ± 0.25 | 0.31 ± 0.007 |
| ResNet-50 | SWAG | 90.64 ± 0.16 | **1.90** ± 0.23 | **0.29** ± 0.003 |
| VGG16 | MAP | 93.02 ± 0.19 | 4.88 ± 0.26 | 0.34 ± 0.010 |
| VGG16 | MC-drop | 93.02 ± 0.19 | 4.44 ± 0.26 | 0.31 ± 0.010 |
| VGG16 | LA | 93.04 ± 0.16 | 2.66 ± 0.23 | 0.25 ± 0.006 |
| VGG16 | a-GPS-4 | 91.64 ± 0.12 | 43.60 ± 4.96 | 0.85 ± 0.103 |
| VGG16 | a-GPS-4-SWA | 39.58 ± 18.62 | 60.34 ± 18.72 | 9.63 ± 3.010 |
| VGG16 | a-GPS-4-SWAG | 45.06 ± 18.63 | 25.02 ± 5.72 | 2.08 ± 0.762 |
| VGG16 | a-GPS-5 | 92.42 ± 0.23 | 5.78 ± 0.29 | 0.48 ± 0.017 |
| VGG16 | a-GPS-5-SWA | 92.48 ± 0.23 | 7.52 ± 0.23 | 1.20 ± 0.039 |
| VGG16 | a-GPS-5-SWAG | 92.32 ± 0.20 | 2.16 ± 0.19 | 0.37 ± 0.031 |
| VGG16 | a-GPS-6 | 93.04 ± 0.15 | 6.08 ± 0.21 | 0.46 ± 0.015 |
| VGG16 | a-GPS-6-SWA | 93.04 ± 0.15 | 6.38 ± 0.21 | 0.75 ± 0.030 |
| VGG16 | a-GPS-6-SWAG | 93.06 ± 0.15 | 6.20 ± 0.20 | 0.45 ± 0.012 |
| VGG16 | a-GPS-7 | 93.04 ± 0.21 | 4.96 ± 0.22 | 0.35 ± 0.011 |
| VGG16 | a-GPS-7-SWA | 93.06 ± 0.17 | 4.96 ± 0.22 | 0.35 ± 0.011 |
| VGG16 | a-GPS-7-SWAG | 93.04 ± 0.21 | 4.96 ± 0.22 | 0.35 ± 0.011 |
| VGG16 | SGD-MC | **93.22** ± 0.23 | 1.30 ± 0.24 | **0.21** ± 0.004 |
| VGG16 | SWA | 93.18 ± 0.16 | 4.18 ± 0.15 | 0.27 ± 0.005 |
| VGG16 | SWAG | 93.20 ± 0.18 | **1.08** ± 0.24 | **0.21** ± 0.003 |
| WideResNet28-10 | MAP | 96.22 ± 0.21 | 2.08 ± 0.20 | 0.14 ± 0.009 |
| WideResNet28-10 | LA | 96.20 ± 0.27 | **0.96** ± 0.14 | 0.13 ± 0.007 |
| WideResNet28-10 | a-GPS-4 | 95.08 ± 0.22 | 2.42 ± 0.17 | 0.17 ± 0.007 |
| WideResNet28-10 | a-GPS-4-SWA | 94.94 ± 0.21 | 3.74 ± 0.19 | 0.27 ± 0.014 |
| WideResNet28-10 | a-GPS-4-SWAG | 94.10 ± 0.18 | 11.22 ± 1.68 | 0.29 ± 0.020 |
| WideResNet28-10 | a-GPS-5 | 95.94 ± 0.10 | 2.12 ± 0.15 | 0.15 ± 0.008 |
| WideResNet28-10 | a-GPS-5-SWA | 95.80 ± 0.18 | 2.56 ± 0.14 | 0.17 ± 0.008 |
| WideResNet28-10 | a-GPS-5-SWAG | 95.78 ± 0.12 | 2.94 ± 0.16 | 0.17 ± 0.005 |
| WideResNet28-10 | a-GPS-6 | 96.20 ± 0.22 | 2.10 ± 0.17 | 0.15 ± 0.008 |
| WideResNet28-10 | a-GPS-6-SWA | 96.18 ± 0.19 | 2.22 ± 0.15 | 0.15 ± 0.008 |
| WideResNet28-10 | a-GPS-6-SWAG | 96.22 ± 0.18 | 2.12 ± 0.17 | 0.15 ± 0.008 |
| WideResNet28-10 | a-GPS-7 | 96.26 ± 0.26 | 2.04 ± 0.16 | 0.14 ± 0.008 |
| WideResNet28-10 | a-GPS-7-SWA | 96.22 ± 0.26 | 2.16 ± 0.16 | 0.14 ± 0.008 |
| WideResNet28-10 | a-GPS-7-SWAG | 96.22 ± 0.21 | 2.08 ± 0.19 | 0.14 ± 0.008 |
| WideResNet28-10 | SGD-MC | 96.40 ± 0.18 | 3.40 ± 0.19 | 0.13 ± 0.003 |
| WideResNet28-10 | SWA | **96.66** ± 0.12 | 1.82 ± 0.18 | 0.13 ± 0.006 |
| WideResNet28-10 | SWAG | 96.26 ± 0.08 | 1.66 ± 0.22 | **0.12** ± 0.001 |

Table 8: CIFAR10 prediction time relative to MAP.

| Model | Method | Time ↓ |
|---|---|---|
| ResNet-50 | LA | 0.26 |
| ResNet-50 | MAP | 1.00 |
| ResNet-50 | SWA | 1.19 |
| ResNet-50 | a-GPS | 4.22 |
| ResNet-50 | SWAG | 17.51 |
| VGG16 | LA | 0.16 |
| VGG16 | MAP | 1.00 |
| VGG16 | SWA | 1.09 |
| VGG16 | MC-dropout | 2.06 |
| VGG16 | a-GPS | 2.85 |
| VGG16 | SWAG | 10.91 |
| WideResNet28-10 | LA | 0.52 |
| WideResNet28-10 | MAP | 1.00 |
| WideResNet28-10 | SWA | 1.67 |
| WideResNet28-10 | a-GPS | 11.40 |
| WideResNet28-10 | SWAG | 35.92 |

Table 9: CIFAR100 for three different architectures. Laplace did get nan errors for VGG16.

| Model | Method | Acc ↑ | ECE ↓ | NLL ↓ |
|---|---|---|---|---|
| ResNet-50 | MAP | 58.45 ± 1.40 | 17.00 ± 0.52 | **1.84** ± 0.054 |
| ResNet-50 | LA | 57.78 ± 1.34 | 12.88 ± 0.37 | 1.84 ± 0.055 |
| ResNet-50 | a-GPS-4 | 54.40 ± 1.62 | 28.40 ± 0.88 | 2.61 ± 0.083 |
| ResNet-50 | a-GPS-4-SWA | 53.68 ± 1.59 | 42.80 ± 1.60 | 6.16 ± 0.273 |
| ResNet-50 | a-GPS-4-SWAG | 54.76 ± 1.48 | 14.88 ± 1.74 | 2.24 ± 0.170 |
| ResNet-50 | a-GPS-5 | 57.18 ± 1.64 | 24.44 ± 0.94 | 2.17 ± 0.077 |
| ResNet-50 | a-GPS-5-SWA | 57.06 ± 1.64 | 27.14 ± 1.09 | 2.47 ± 0.115 |
| ResNet-50 | a-GPS-5-SWAG | 57.48 ± 1.60 | 19.80 ± 0.68 | 1.97 ± 0.067 |
| ResNet-50 | a-GPS-6 | 58.00 ± 1.63 | 18.18 ± 0.76 | 1.88 ± 0.061 |
| ResNet-50 | a-GPS-6-SWA | 57.96 ± 1.64 | 18.46 ± 0.77 | 1.90 ± 0.062 |
| ResNet-50 | a-GPS-6-SWAG | 58.06 ± 1.64 | 18.10 ± 0.79 | 1.88 ± 0.061 |
| ResNet-50 | a-GPS-7 | 58.02 ± 1.53 | 17.10 ± 0.52 | 1.86 ± 0.057 |
| ResNet-50 | a-GPS-7-SWA | 57.82 ± 1.62 | 17.40 ± 0.61 | 1.87 ± 0.055 |
| ResNet-50 | a-GPS-7-SWAG | 57.96 ± 1.56 | 17.10 ± 0.52 | 1.86 ± 0.055 |
| ResNet-50 | SGD-MC | **59.78** ± 1.42 | 2.12 ± 0.35 | 1.56 ± 0.047 |
| ResNet-50 | SWA | 59.26 ± 1.36 | 19.68 ± 1.07 | **1.84** ± 0.074 |
| ResNet-50 | SWAG | 59.58 ± 1.26 | **2.00** ± 0.47 | 1.55 ± 0.047 |
| VGG16 | MAP | 70.86 ± 0.37 | 20.74 ± 0.30 | 1.96 ± 0.017 |
| VGG16 | LA | nan | nan | nan |
| VGG16 | MC-drop | 70.78 ± 0.35 | 16.74 ± 0.29 | 1.63 ± 0.010 |
| VGG16 | a-GPS-4 | 68.78 ± 0.19 | 44.50 ± 2.97 | 2.15 ± 0.151 |
| VGG16 | a-GPS-4-SWA | 6.50 ± 4.32 | 11.32 ± 7.33 | 5.61 ± 0.972 |
| VGG16 | a-GPS-4-SWAG | 28.28 ± 16.38 | 18.68 ± 10.97 | 3.90 ± 0.621 |
| VGG16 | a-GPS-5 | 69.46 ± 0.26 | 23.12 ± 0.24 | 2.56 ± 0.013 |
| VGG16 | a-GPS-5-SWA | 69.32 ± 0.25 | 30.40 ± 0.26 | 4.81 ± 0.045 |
| VGG16 | a-GPS-5-SWAG | 69.70 ± 0.17 | 19.40 ± 0.18 | 2.55 ± 0.021 |
| VGG16 | a-GPS-6 | 70.96 ± 0.42 | 22.28 ± 0.42 | 2.23 ± 0.017 |
| VGG16 | a-GPS-6-SWA | 70.96 ± 0.42 | 22.46 ± 0.40 | 2.32 ± 0.019 |
| VGG16 | a-GPS-6-SWAG | 70.92 ± 0.44 | 22.38 ± 0.42 | 2.28 ± 0.021 |
| VGG16 | a-GPS-7 | 70.84 ± 0.36 | 20.90 ± 0.28 | 1.98 ± 0.014 |
| VGG16 | a-GPS-7-SWA | 70.84 ± 0.36 | 20.92 ± 0.29 | 1.98 ± 0.014 |
| VGG16 | a-GPS-7-SWAG | 70.84 ± 0.36 | 20.90 ± 0.30 | 1.98 ± 0.014 |
| VGG16 | SGD-MC | **72.58** ± 0.22 | **1.32** ± 0.10 | **1.02** ± 0.012 |
| VGG16 | SWA | 71.98 ± 0.32 | 16.48 ± 0.28 | 1.40 ± 0.032 |
| VGG16 | SWAG | 72.32 ± 0.23 | 1.42 ± 0.31 | 1.03 ± 0.017 |
| WRN28-10 | MAP | 79.50 ± 0.30 | 5.94 ± 0.27 | 0.89 ± 0.013 |
| WRN28-10 | LA | 79.40 ± 0.35 | 18.52 ± 0.65 | 1.02 ± 0.015 |
| WRN28-10 | a-GPS-4 | 78.64 ± 0.26 | **2.60** ± 0.33 | 0.78 ± 0.013 |
| WRN28-10 | a-GPS-4-SWA | 77.70 ± 0.28 | 9.56 ± 0.70 | 0.97 ± 0.024 |
| WRN28-10 | a-GPS-4-SWAG | 74.92 ± 0.15 | 46.76 ± 1.48 | 1.72 ± 0.050 |
| WRN28-10 | a-GPS-5 | 79.82 ± 0.31 | 3.92 ± 0.28 | 0.82 ± 0.013 |
| WRN28-10 | a-GPS-5-SWA | 79.64 ± 0.37 | 4.82 ± 0.22 | 0.84 ± 0.014 |
| WRN28-10 | a-GPS-5-SWAG | 79.90 ± 0.22 | 5.08 ± 0.27 | 0.83 ± 0.014 |
| WRN28-10 | a-GPS-6 | 79.64 ± 0.33 | 6.30 ± 0.36 | 0.88 ± 0.016 |
| WRN28-10 | a-GPS-6-SWA | 79.68 ± 0.35 | 7.28 ± 0.45 | 0.91 ± 0.016 |
| WRN28-10 | a-GPS-6-SWAG | 79.72 ± 0.26 | 7.28 ± 0.37 | 0.90 ± 0.016 |
| WRN28-10 | a-GPS-7 | 79.86 ± 0.36 | 5.40 ± 0.19 | 0.86 ± 0.014 |
| WRN28-10 | a-GPS-7-SWA | 79.76 ± 0.30 | 6.14 ± 0.35 | 0.88 ± 0.015 |
| WRN28-10 | a-GPS-7-SWAG | 79.84 ± 0.29 | 6.12 ± 0.32 | 0.87 ± 0.014 |
| WRN28-10 | SGD-MC | 80.44 ± 0.24 | 9.34 ± 0.25 | 0.76 ± 0.007 |
| WRN28-10 | SWA | **80.64** ± 0.27 | 6.76 ± 0.08 | 0.77 ± 0.011 |
| WRN28-10 | SWAG | 80.14 ± 0.21 | 4.80 ± 0.60 | **0.73** ± 0.007 |

Table 10: CIFAR100 prediction time relative to MAP.

| Model | Method | Time ↓ |
|---|---|---|
| ResNet-50 | MAP | 1.00 |
| ResNet-50 | SWA | 1.20 |
| ResNet-50 | LA | 3.01 |
| ResNet-50 | a-GPS | 4.30 |
| ResNet-50 | SWAG | 17.07 |
| VGG16 | MAP | 1.00 |
| VGG16 | SWA | 1.12 |
| VGG16 | MC-dropout | 2.16 |
| VGG16 | a-GPS | 2.89 |
| VGG16 | SWAG | 10.80 |
| VGG16 | LA | nan |
| WideResNet28-10 | MAP | 1.00 |
| WideResNet28-10 | SWA | 1.50 |
| WideResNet28-10 | a-GPS | 10.42 |
| WideResNet28-10 | LA | 17.93 |
| WideResNet28-10 | SWAG | 32.24 |

Table 11: Predictive entropies for the CIFAR5-5 in (IND) and out (OOD) of distribution. For IND lower is better, for OOD higher is better.

| Model | Method | IND ENT ↓ | OOD ENT ↑ |
|---|---|---|---|
| ResNet-50 | MAP | 0.13 ± 0.006 | 0.46 ± 0.007 |
| ResNet-50 | LA | 0.18 ± 0.004 | 0.46 ± 0.007 |
| ResNet-50 | a-GPS-4 | 0.11 ± 0.006 | 0.41 ± 0.021 |
| ResNet-50 | a-GPS-4-SWA | **0.04** ± 0.003 | 0.09 ± 0.010 |
| ResNet-50 | a-GPS-4-SWAG | 0.35 ± 0.013 | 0.71 ± 0.011 |
| ResNet-50 | a-GPS-5 | 0.09 ± 0.004 | 0.34 ± 0.012 |
| ResNet-50 | a-GPS-5-SWA | 0.08 ± 0.004 | 0.18 ± 0.008 |
| ResNet-50 | a-GPS-5-SWAG | 0.09 ± 0.003 | 0.24 ± 0.010 |
| ResNet-50 | a-GPS-6 | 0.12 ± 0.005 | 0.43 ± 0.010 |
| ResNet-50 | a-GPS-6-SWA | 0.12 ± 0.005 | 0.25 ± 0.004 |
| ResNet-50 | a-GPS-6-SWAG | 0.12 ± 0.005 | 0.26 ± 0.004 |
| ResNet-50 | a-GPS-7 | 0.13 ± 0.005 | 0.45 ± 0.010 |
| ResNet-50 | a-GPS-7-SWA | 0.12 ± 0.005 | 0.26 ± 0.005 |
| ResNet-50 | a-GPS-7-SWAG | 0.13 ± 0.005 | 0.27 ± 0.004 |
| ResNet-50 | SGD-MC | 0.43 ± 0.009 | **1.18** ± 0.013 |
| ResNet-50 | SWA | 0.19 ± 0.004 | 0.36 ± 0.005 |
| ResNet-50 | SWAG | 0.38 ± 0.008 | 0.99 ± 0.016 |
| VGG16 | MAP | 0.10 ± 0.003 | 0.53 ± 0.022 |
| VGG16 | MC-drop | 0.11 ± 0.003 | 0.61 ± 0.023 |
| VGG16 | LA | 0.35 ± 0.010 | 0.53 ± 0.022 |
| VGG16 | a-GPS-4 | 0.13 ± 0.009 | 0.61 ± 0.058 |
| VGG16 | a-GPS-4-SWA | nan ± nan | nan ± nan |
| VGG16 | a-GPS-5 | 0.06 ± 0.003 | 0.26 ± 0.014 |
| VGG16 | a-GPS-5-SWA | nan ± nan | nan ± nan |
| VGG16 | a-GPS-6 | 0.05 ± 0.002 | 0.27 ± 0.014 |
| VGG16 | a-GPS-6-SWA | **0.04** ± 0.002 | 0.20 ± 0.012 |
| VGG16 | a-GPS-6-SWAG | **0.04** ± 0.002 | 0.25 ± 0.013 |
| VGG16 | a-GPS-7 | 0.09 ± 0.003 | 0.50 ± 0.021 |
| VGG16 | a-GPS-7-SWA | 0.09 ± 0.003 | 0.50 ± 0.021 |
| VGG16 | a-GPS-7-SWAG | 0.09 ± 0.003 | 0.50 ± 0.021 |
| VGG16 | SGD-MC | 0.29 ± 0.004 | **1.13** ± 0.027 |
| VGG16 | SWA | 0.11 ± 0.004 | 0.59 ± 0.023 |
| VGG16 | SWAG | 0.29 ± 0.006 | 1.12 ± 0.037 |
| WideResNet28-10 | MAP | 0.08 ± 0.003 | 0.67 ± 0.016 |
| WideResNet28-10 | LA | 0.12 ± 0.004 | 0.67 ± 0.016 |
| WideResNet28-10 | a-GPS-4 | 0.08 ± 0.005 | 0.60 ± 0.016 |
| WideResNet28-10 | a-GPS-4-SWA | **0.04** ± 0.002 | 0.26 ± 0.004 |
| WideResNet28-10 | a-GPS-4-SWAG | 0.40 ± 0.008 | **0.98** ± 0.021 |
| WideResNet28-10 | a-GPS-5 | 0.07 ± 0.003 | 0.64 ± 0.012 |
| WideResNet28-10 | a-GPS-5-SWA | 0.07 ± 0.003 | 0.46 ± 0.006 |
| WideResNet28-10 | a-GPS-5-SWAG | 0.19 ± 0.013 | 0.67 ± 0.012 |
| WideResNet28-10 | a-GPS-6 | 0.07 ± 0.003 | 0.66 ± 0.012 |
| WideResNet28-10 | a-GPS-6-SWA | 0.07 ± 0.003 | 0.52 ± 0.006 |
| WideResNet28-10 | a-GPS-6-SWAG | 0.08 ± 0.003 | 0.53 ± 0.006 |
| WideResNet28-10 | a-GPS-7 | 0.08 ± 0.003 | 0.68 ± 0.018 |
| WideResNet28-10 | a-GPS-7-SWA | 0.08 ± 0.003 | 0.54 ± 0.007 |
| WideResNet28-10 | a-GPS-7-SWAG | 0.08 ± 0.003 | 0.54 ± 0.007 |
| WideResNet28-10 | SGD-MC | 0.13 ± 0.059 | 0.85 ± 0.151 |
| WideResNet28-10 | SWA | 0.08 ± 0.003 | 0.54 ± 0.008 |
| WideResNet28-10 | SWAG | 0.12 ± 0.039 | 0.77 ± 0.186 |

Table 12: Predictive entropies for the CIFAR50-50 in (IND) and out (OOD) of distribution. For IND lower is better, for OOD higher is better.

| Model | Method | IND ENT ↓ | OOD ENT ↑ |
|---|---|---|---|
| ResNet-50 | MAP | 1.01 ± 0.120 | 1.98 ± 0.164 |
| ResNet-50 | LA | 3.91 ± 0.092 | 2.21 ± 0.159 |
| ResNet-50 | a-GPS-4 | 0.50 ± 0.034 | 0.91 ± 0.044 |
| ResNet-50 | a-GPS-4-SWA | **0.15** ± 0.016 | 0.22 ± 0.017 |
| ResNet-50 | a-GPS-4-SWAG | 0.95 ± 0.061 | 1.66 ± 0.060 |
| ResNet-50 | a-GPS-5 | 0.67 ± 0.068 | 1.36 ± 0.099 |
| ResNet-50 | a-GPS-5-SWA | 0.58 ± 0.060 | 0.87 ± 0.027 |
| ResNet-50 | a-GPS-5-SWAG | 0.72 ± 0.076 | 1.13 ± 0.045 |
| ResNet-50 | a-GPS-6 | 0.96 ± 0.104 | 1.89 ± 0.144 |
| ResNet-50 | a-GPS-6-SWA | 0.94 ± 0.100 | 1.43 ± 0.060 |
| ResNet-50 | a-GPS-6-SWAG | 0.96 ± 0.106 | 1.46 ± 0.057 |
| ResNet-50 | a-GPS-7 | 1.02 ± 0.111 | 1.99 ± 0.150 |
| ResNet-50 | a-GPS-7-SWA | 1.01 ± 0.108 | 1.54 ± 0.068 |
| ResNet-50 | a-GPS-7-SWAG | 1.03 ± 0.112 | 1.56 ± 0.064 |
| ResNet-50 | SGD-MC | 1.31 ± 0.132 | **2.38** ± 0.140 |
| ResNet-50 | SWA | 1.06 ± 0.131 | 1.59 ± 0.100 |
| ResNet-50 | SWAG | 1.31 ± 0.119 | 2.03 ± 0.204 |
| VGG16 | MAP | nan ± nan | 0.85 ± 0.008 |
| VGG16 | MC-drop | **0.46** ± 0.006 | 1.17 ± 0.012 |
| VGG16 | a-GPS-4 | nan ± nan | 0.83 ± 0.008 |
| VGG16 | a-GPS-4-SWA | nan ± nan | 0.83 ± 0.008 |
| VGG16 | a-GPS-5 | nan ± nan | nan ± nan |
| VGG16 | a-GPS-5-SWA | nan ± nan | 0.39 ± 0.009 |
| VGG16 | a-GPS-6 | nan ± nan | 0.94 ± 0.010 |
| VGG16 | a-GPS-6-SWA | nan ± nan | 0.67 ± 0.007 |
| VGG16 | a-GPS-7 | nan ± nan | nan ± nan |
| VGG16 | a-GPS-7-SWA | nan ± nan | 0.83 ± 0.008 |
| VGG16 | SGD | 0.91 ± 0.004 | 1.03 ± 0.410 |
| VGG16 | SWA | nan ± nan | nan ± nan |
| VGG16 | SWAG | 0.85 ± 0.011 | **2.00** ± 0.017 |
| WideResNet28-10 | MAP | 0.98 ± 0.105 | 1.08 ± 0.011 |
| WideResNet28-10 | LA | 2.99 ± 0.060 | 1.91 ± 0.015 |
| WideResNet28-10 | a-GPS-4 | 0.64 ± 0.058 | 2.28 ± 0.119 |
| WideResNet28-10 | a-GPS-4-SWA | **0.30** ± 0.010 | 2.26 ± 0.129 |
| WideResNet28-10 | a-GPS-4-SWAG | 2.38 ± 0.355 | 1.82 ± 0.108 |
| WideResNet28-10 | a-GPS-5 | 0.85 ± 0.060 | 2.38 ± 0.127 |
| WideResNet28-10 | a-GPS-5-SWA | 0.77 ± 0.053 | 2.38 ± 0.141 |
| WideResNet28-10 | a-GPS-5-SWAG | 0.86 ± 0.055 | 2.69 ± 0.113 |
| WideResNet28-10 | a-GPS-6 | 1.04 ± 0.102 | 2.95 ± 0.189 |
| WideResNet28-10 | a-GPS-6-SWA | 1.04 ± 0.107 | 3.07 ± 0.182 |
| WideResNet28-10 | a-GPS-6-SWAG | 1.05 ± 0.105 | 3.08 ± 0.189 |
| WideResNet28-10 | a-GPS-7 | 1.00 ± 0.100 | 2.11 ± 0.108 |
| WideResNet28-10 | a-GPS-7-SWA | 1.02 ± 0.107 | 2.31 ± 0.131 |
| WideResNet28-10 | a-GPS-7-SWAG | 1.02 ± 0.104 | 3.02 ± 0.192 |
| WideResNet28-10 | SGD-MC | 1.38 ± 0.142 | 1.04 ± 0.022 |
| WideResNet28-10 | SWA | 0.67 ± 0.045 | **3.42** ± 0.318 |
| WideResNet28-10 | SWAG | 1.28 ± 0.171 | 3.17 ± 0.112 |

Table 13: Results Imagenet with ResNet-50 architecture.

| Method | Acc ↑ | ECE ↓ | NLL ↓ |
|---|---|---|---|
| MAP | 76.10 ± 0.06 | 3.34 ± 0.08 | 0.95 ± 0.004 |
| LA | 75.86 ± 0.10 | 15.32 ± 0.12 | 1.05 ± 0.004 |
| a-GPS-4 | 70.74 ± 0.12 | 8.14 ± 0.15 | 1.25 ± 0.004 |
| a-GPS-1-4 | 73.98 ± 0.10 | 5.04 ± 0.05 | 1.08 ± 0.006 |
| a-GPS-1-4-SWA | 66.56 ± 0.85 | 6.86 ± 0.26 | 1.62 ± 0.053 |
| a-GPS-1-4-SWAG | 62.42 ± 0.37 | 42.82 ± 0.36 | 2.53 ± 0.015 |
| a-GPS-5 | 75.76 ± 0.10 | 3.82 ± 0.07 | 0.98 ± 0.004 |
| a-GPS-1-5 | 76.02 ± 0.10 | 4.54 ± 0.05 | 0.97 ± 0.004 |
| a-GPS-1-5-SWA | 75.94 ± 0.08 | 5.46 ± 0.05 | 0.99 ± 0.004 |
| a-GPS-1-5-SWAG | 76.00 ± 0.10 | 3.73 ± 0.11 | 0.96 ± 0.004 |
| a-GPS-6 | 76.14 ± 0.08 | 3.70 ± 0.09 | 0.95 ± 0.004 |
| a-GPS-1-6 | 76.12 ± 0.07 | 3.60 ± 0.00 | 0.95 ± 0.004 |
| a-GPS-1-6-SWA | 76.10 ± 0.11 | 3.66 ± 0.05 | 0.95 ± 0.004 |
| a-GPS-1-6-SWAG | 76.07 ± 0.04 | 3.68 ± 0.04 | 0.95 ± 0.004 |
| SGD-MC | **76.58** ± 0.13 | **1.88** ± 0.04 | **0.91** ± 0.004 |
| SWA | 76.45 ± 0.05 | 2.15 ± 0.05 | 0.93 ± 0.002 |
| SWAG | 76.50 ± 0.10 | 5.05 ± 0.15 | 0.94 ± 0.001 |

