# OpenReview forum: "Exploring Deep Learning Parameter Space with a-GPS: Approximate Gaussian Proposal Sampler"
_ICLR.cc/2024/Conference — Submitted to ICLR 2024_

### Official Review · Reviewer_XKu5 · 2023-10-26

**Soundness:** 1 poor
**Presentation:** 1 poor
**Contribution:** 1 poor
**Rating:** 1
**Confidence:** 3

**Summary:**

The paper proposes a method to obtain Gaussian approximations of posterior distributions in Bayesian deep learning. The experiments compare the proposed method against several related approaches on toy experiments as well as classification on CIFAR-10/100 and ImageNet.

**Strengths:**

The authors report that their method tends to produce samples quicker than competitor methods.

**Weaknesses:**

The paper is definitely still a work in progress and not ready for publication at a conference like ICLR.
Thus, I vote for rejection and encourage the authors to completely revise their manuscript and submit to another venue.

The writing style and organization of the paper is very bad, which makes it extremely hard to follow. In particular, the theoretical exposition is lacking:
- The theory is mixed with the related work (Eqs. (1)-(3), last Sec. of 1.1)
- Central notions and symbols are not introduced, the exposition remains very handwavy. To name only a few examples:
  - what do the authors mean by "transforming a pretrained into a Bayesian model"?
  - background on MCMC, Metropolis-Hastings corrections
  - definition of a "perfect sampler"
  - how do the authors define a "mode-specific MH"
  - it remains unclear in which sense the proposed method better deals with multi-modal posteriors than related work
  - definition of notion of time step $t$ and $\theta_t$ in Eq. (4)
  - definition of $D_x$, $D_y$ in Eq. (15, 16)
  - definition of $\mathrm{Conf}$ in Eq. (20)
  - ...
- The experimental evaluation is not convincing.
  - While the authors report fast sampling, their approach is outperformed by competitor methods most of the time.
  - On the simplest toy example (unimodal Gaussian posterior), the authors report good results in terms of effective sample size (which is not very surprising because they use the correct approximation). However, they do not report ESS on the mixture model (Figure 2 RHS).
  - The authors argue that their method deals well with multi-modal posteriors. Thus, they should compare
 against other methods that capture multiple modes, i.p., Deep Ensembles [1] and Multi-SWAG [2].
  - As the authors employ a Gaussian posterior approximations, they should compare against variational Gaussian approximations, e.g., BayesByBackprop [3].

[1] Lakshminarayanan et al., "Simple and Scalable Predictive Uncertainty Estimation using Deep Ensembles", NeurIPS 2017

[2] Wilson & Izmailov, "Bayesian Deep Learning and a Probabilistic Perspective of Generalization", NeurIPS 2020

[3] Blundell et al., "Weight Uncertainty in Neural Networks", ICML 2015

**Questions:**

Please elaborate on the concerns raised below "Weaknesses".

---

> ### Author Response · Authors · 2023-11-23
> **Response to XKu5**
>
> **Comment 1:**
> *The paper is definitely still a work in progress and not ready for publication at a conference like ICLR. Thus, I vote for rejection and encourage the authors to completely revise their manuscript and submit to another venue.
> The writing style and organization of the paper is very bad, which makes it extremely hard to follow. In particular, the theoretical exposition is lacking.*
>
> *The theory is mixed with the related work (Eqs. (1)-(3), last Sec. of 1.1). Central notions and symbols are not introduced, the exposition remains very handwavy.
> To name only a few examples: what do the authors mean by "transforming a pretrained into a Bayesian model"?, background on MCMC, Metropolis-Hastings corrections, definition of a "perfect sampler", how do the authors define a "mode-specific MH", it remains unclear in which sense the proposed method better deals with multi-modal posteriors (...)*
>
> **Answer 1:**
> We appreciate the valuable feedback from the reviewers, and we believe that the revisions made to the manuscript improve the work considerably to the point that it is in fact ready for publication.
>
> **We have moved the theory of previous work** into the subsection of “Problem setup”; **we have revised and included a description of symbols** in the “Bayesian Setting and Deep Learning” in Appendix A; **we changed the wording of “transforming a pretrained model into a Bayesian model” to
> >“methods for converting pretrained models into quantifiable uncertainty models”**
>
> (paragraph 1, page 1);
>
> We specify a perfect sampler in our revision. In “Related work” paragraph 2 it now reads:> “[...] a perfect sampler, meaning that it must produce independent samples from the true distribution,”
>  and in paragraph 6 on page 2:
>  >“ A perfect sampler q generates a sample \theta with the exact probability of f (θ) (Propp & Wilson, 1996).”
>
>
> The mode-specific MH is now defined more explicitly in eq. 15. Unfortunately, we did not manage to fit in a background regarding MCMC and Metropolis-Hastings, though we do cite the original works. The concern regarding the definition of the notion of a time step in Eq. (4) (now Eq. 1) has been addressed :
> >”To sequentially sample from the posterior, we assume access to theta_t”
>
> Definitions for Dx, Dy, and Conf are properly introduced (now in Appendix H.3.1).
> Spelling errors and missing definitions have been corrected.
>
> We hope the manuscript now aligns more closely with the review’s expectations.
>
> **Comment 2:** *it remains unclear in which sense the proposed method better deals with multi-modal posteriors than related work.*
>
> **Answer 2:**
> We acknowledge the concern, and hope that our revision better communicates the efficiency compared to typical (SG-)MCMC methods with the simulated examples in the related work :
> >§2 “SG-MCMC methods rely on inefficient proposal dynamics that generates few effective samples per epoch, even from a warm-start, making it expensive for deep learning problems.”
>
> >§3 “Our framework addresses these challenges by self-adjusting towards a perfect sampler, meaning that it produces independent samples from the true distribution, obviating the need for an MH correction. The method provides efficient exploration and computation, crucial for dealing with large models and datasets.”
>
> Additional modifications in the last paragraph of the 'Related Work' now highlight that
> LA, SWA, and SWAG face limitations :
> >”  [These] suffer limitations in terms of inexactness and slow exploration, restrictions to a unimodal posterior, reliance on specific model architectures, or computational inefficiency during inference.”

---

> ### Author Response · Authors · 2023-11-23
> **Response to XKu5**
>
> **Comment 3:**
> *The experimental evaluation is not convincing: While the authors report fast sampling, their approach is outperformed by competitor methods most of the time.*
>
> **Answer 3: **
> We agree with the reviewer that the method does not surpass the baselines on standard performance metrics. However, we still believe the method is a substantial contribution as it is **a novel sampler that quickly produces a posterior approximation for any model, it can be multimodal, with high effective sample size within modes**. In contrast to our model, SWA, SWAG and LA may be limited by their Gaussian posterior approximation, while SG-MCMC methods may be limited by their inefficient exploration. This has now been added both in the contributions, results, discussion and conclusion, paragraph 1, 3 and 4 on page 2, paragraph 3 on page 5 and paragraph 3 on page 8.
>
> **Even if statistical novel models don't outperform existing standards, their competitiveness makes them worthy of exploration and publication. By offering unique approaches, these models contribute valuable insights to the field.** We agree with the reviewer that this should have been explicitly stated. Paragraph 2 on page 9 now reads:
> > "While not surpassing standards universally, the model showcases competitiveness as a novel sampler, swiftly generating a posterior approximation for any model. Its multimodal capability, coupled with a high effective sample size within modes, underscores the need for further exploration and elaboration.”
>
> We thank the reviewer for pointing towards the area where it becomes obvious that we did not manage to convey a clear and concise message. **Our contribution is a novel sampler that quickly produces a posterior approximation for any model, it can be multimodal, with high effective sample size within modes.** In contrast to SWAG which is limited by its Gaussian posterior approximation. SGD-MC may also be limited to one mode as it records the trace of an optimization method, and suffers from a low effective sample size. **We now also include Langevin dynamics in the simulated experiments** as this was requested by multiple reviewers. See table below.
>
> **Comment 4:**
> *On the simplest toy example (unimodal Gaussian posterior), the authors report good results in terms of effective sample size (which is not very surprising because they use the correct approximation). However, they do not report ESS on the mixture model (Figure 2 RHS).*
>
> **Answer 4:**
> We are grateful for the opportunity to clarify this. Though the unimodal Gaussian example may seem simple, it's worth noting that the two other samplers HMC and Langevin dynamics struggle to produce good ESS for this simple case. We did not originally report the ESS on the mixture model as the ESS assumes one mode. However, we have now calculated and included it to the results in the revised version the table is
> | Method      | Unimodal ESS | Unimodal Time (s) | Mixture ESS | Mixture Time (s) |
> |-------------|--------------|-------------------|-------------|-------------------|
> | **a-GPS**    | 954          | 0.4               | 70.2        | 2                 |
> | **HMC**      | 364          | 4.6               | 105.2       | 30                |
> | **Langevin** | 74           | 2.3               | 36.8        | 4.9               |
>
>
>
> **Comment 5:**
> *The authors argue that their method deals well with multi-modal posteriors. Thus, they should compare against other methods that capture multiple modes, i.p., Deep Ensembles [1] and Multi-SWAG [2].*
>
> **Answer 5:**
> We agree with the reviewer, that it would be intriguing to explore in future work how our methods perform when multiple pretrained models are employed, as observed in Deep Ensembles and Multi-SWAG. However, **we respectfully disagree with considering these methods as relevant comparisons to the proposed method, given their reliance on multiple pretrained networks.** Our paper specifically concentrates on scenarios where a single pretrained model serves as the foundation for generating new models. This is now made explicit in the first paragraph under “Deep Learning Experiments” :
> > “All methods utilize a single pretrained network as a warm start”.
>
> **Comment 6:**
> *As the authors employ a Gaussian posterior approximations, they should compare against variational Gaussian approximations, e.g., BayesByBackprop [3].*
>
> **Answer 6:**
> While we appreciate the suggestion, the paper focuses on the context of a pretrained model. Consequently, we deemed it more relevant to compare with Laplace approximation, considering it a close relative to VI.

---

### Official Review · Reviewer_xaq1 · 2023-10-29

**Soundness:** 2 fair
**Presentation:** 2 fair
**Contribution:** 2 fair
**Rating:** 3
**Confidence:** 5

**Summary:**

This paper proposes an adaptive proposal sampling (APS), a mode seeking sampler that adapts the proposal to match a posterior mode.

**Strengths:**

The proposed ``adaptive proposal sampler'' appears to be new in the literature.

**Weaknesses:**

1. Extension of the proposed sampler to high-dimensional problems is questionable. As mentioned in the paper, the parameters are regarded as independent of each other, making the proposed sampler less accurate and thus less attractive.

2. When the modes of the target distribution are well separated, it is difficult to believe that the proposed sampler can efficiently traverse the entire energy landscape because, similar to the Metropolis-Hastings algorithm, the proposed sampler lacks a mode-escaping mechanism.

3. For the exact Gaussian proposal sampler, the acceptance rate can be low when the dimension of \theta is high.

**Questions:**

1. If the exact GPS is applied to the numerical examples of the paper, will the reported results be improved? How much?

2. The proposed method needs to compare with more baseline methods, such as SGHMC [1]  and adaptively weighted SGLD [2], on multi-modal and high-dimensional problems.

References:

[1] Chen et al. (2014) Stochastic Gradient Hamiltonian Monte Carlo. ICML 2014.

[2]  Deng et al. (2022) An adaptively weighted stochastic gradient MCMC algorithm
for Monte Carlo simulation and global optimization. Statistics and Computing, 32:58.

---

> ### Author Response · Authors · 2023-11-23
> **Response to xaq1**
>
> **Comment 1:**
> *Extension of the proposed sampler to high-dimensional problems is questionable. As mentioned in the paper, the parameters are regarded as independent of each other, making the proposed sampler less accurate and thus less attractive.*
>
> **Answer 1:**
> When it comes to the assumption of independent parameters, this is a common assumption most methods have to make either for computational benefits or memory limits. This assumption is also made by SGHMC, SGLD, LA-diag, SWA, SWAG-diag and SGD-MC.
>
> **Comment 2:**
> *When the modes of the target distribution are well separated, it is difficult to believe that the proposed sampler can efficiently traverse the entire energy landscape because, similar to the Metropolis-Hastings algorithm, the proposed sampler lacks a mode-escaping mechanism.*
>
> **Answer 2:**
> The problem of mode-escaping is a problem in MCMC. A common solution involves using temperature scaling (Neal, 1996)[1] to flatten the modes, thereby increasing the chance of mode overlap. While temperature scaling could be an extension applied to our method for mode-escaping, it is unnecessary in deep learning experiments due to substantial mode overlap already present (Draxler et al., 2019; Garipov et al., 2018).
>
> **Comment 3:**
> *For the exact Gaussian proposal sampler, the acceptance rate can be low when the dimension of \theta is high.*
>
> **Answer 3:**
> The curse of dimensionality is a known problem in general and especially in MCMC. However, **we consider tackling this broad issue as beyond the scope of the paper**. Our work is primarily concerned with proposing a novel sampler that quickly produces a posterior approximation for any model with high effective sample size.
>
> ## Questions
>
> **Question 1:**
> *If the exact GPS is applied to the numerical examples of the paper, will the reported results be improved? How much?*
>
> **Answer 1:**
> While the application of the exact GPS in deep learning might yield better performance, as the Metropolis-Hastings correction is likely to reject models with higher loss, it's crucial to note that the MH correction is computationally expensive (effectively intractable). In deep learning, it is often avoided or approximated due to these computational challenges. We do not know about the possible improvement but we refer to Draxler et al.’s [1] work, who report similar results for full-gradient HMC and SG-HMC.
>
> **Question 2:**
> *The proposed method needs to compare with more baseline methods, such as SGHMC [1] and adaptively weighted SGLD [2], on multi-modal and high-dimensional problems.*
>
> **Answer 2:**
> We are grateful for this suggestion. We opted not to compare with SGLD or SGHMC, because these models are outperformed by LA, SWA, and SWAG on deep learning [5]. Moreover, for the specific task of converting an existing pretrained deep learning model into a Bayesian network, the methods we compared to do not necessitate an extensive burn-in/training time and hyperparameter search, unlike SGHMC and SGLD. However, we do include comparisons with HMC and Langevin dynamics, which serve as typical sampling baselines, in the simulated examples. The table of simulated experiments report :
> | Method      | Unimodal ESS | Unimodal Time (s) | Mixture ESS | Mixture Time (s) |
> |-------------|--------------|-------------------|-------------|-------------------|
> | **a-GPS**    | 954          | 0.4               | 70.2        | 2                 |
> | **HMC**      | 364          | 4.6               | 105.2       | 30                |
> | **Langevin** | 74           | 2.3               | 36.8        | 4.9               |
>
>
>
> [1] Draxler et al., 2019 “Essentially No Barriers in Neural Network Energy Landscape”
>
> [2] Garipov et al., 2018 “Loss Surfaces, Mode Connectivity, and Fast Ensembling of DNNs”
>
> [3] Neal, 1996 “Sampling from multimodal distributions using tempered transitions”
>
> [4] Izmailov et al. 2021 “What Are Bayesian Neural Network Posteriors Really Like?”
>
> [5] Maddox et al., 2019 “A Simple Baseline for Bayesian Uncertainty in Deep Learning”

---

### Official Review · Reviewer_zCTz · 2023-10-29

**Soundness:** 1 poor
**Presentation:** 2 fair
**Contribution:** 2 fair
**Rating:** 3
**Confidence:** 4

**Summary:**

The paper proposes a new sampling algorithm for multi-modal distributions, especially deep neural network posteriors. Specifically, the authors learn an adaptive Gaussian proposal along with sampling. Several experiments, including synthetic distributions and deep learning tasks, are conducted to test the proposed method.

**Strengths:**

1.	The studied topic of sampling on multi-modal distributions is important.
2.	The proposed algorithm is simple to implement in practice.

**Weaknesses:**

1.	The proposed method does not achieve what it claims to “having both exactness and effectiveness”. Apparently, the method is not exact without the MH correction step. The method is only exact when the target distribution is a Gaussian with a diagonal covariance, which is a trivial case. I’m not sure what “perfect sampler” means in the paper. Overall, I think many claims need to be modified in order to be accurate and rigorous.
2.	The methodology of the proposed method is confusing. The algorithm does not have a component to encourage exploring multiple modes. It is unclear to me how the method manages to find diverse modes.
3.	Algorithm 1 seems to find a Gaussian distribution to approximate the target distribution. How is it different from variational inference? What are the advantages?
4.	Why does the proposed method require a pretrained solution, theta_MAP? Will it work if training from scratch?
8.	I do not follow the reason for introducing the variance limit lambda. Why does the method need it?
9.	The experimental setups and results are confusing. It is unclear if the authors also use a pre-trained solution for the baseline NUTS in S3.1. If not, then it is unfair to claim faster convergence of the proposed method than NUTS. Besides, given that the method uses a pre-trained solution, it is unsurprising that “We found that a-GPS converges so fast that a burn-in period was unnecessary”. For the time comparison, it is unclear if the authors include pre-training time.
10.	For deep learning experiments, it will be better to include MCMC baselines, e.g. Zhang et al, as the proposed method belongs to MCMC methods. To show the samples are from diverse modes, the authors can visualize weight space and function space, similar to those in Zhang et al.


Zhang et al, Cyclical Stochastic Gradient MCMC for Bayesian Deep Learning, ICLR 2020

**Questions:**

1.	Why is LA’s inference time even less than MAP? Why is the proposed method’s inference time less than SWAG? Does the proposed method use Bayesian model averaging during inference?

---

> ### Author Response · Authors · 2023-11-23
> **Response to zCTz**
>
> **Comment 1:**
> *The proposed method does not achieve what it claims to “having both exactness and effectiveness”. Apparently, the method is not exact without the MH correction step. The method is only exact when the target distribution is a Gaussian with a diagonal covariance, which is a trivial case. I’m not sure what “perfect sampler” means in the paper. Overall, I think many claims need to be modified in order to be accurate and rigorous.*
>
> **Answer 1:**
> We are grateful for the opportunity to clarify this. **The method maintains precision even without the Metropolis-Hastings correction step, as detailed in Appendix B.**
>
> **For the case where the true distribution is a Gaussian, we achieve what we regard as having both exactness and effectiveness, by both being faster than Langevin dynamics and more exact compared to the gold standard of Hamiltonian Monte Carlo.** We agree with the reviewer in that we may only provide theoretical guarantees for exactness when the target is a diagonal Gaussian distribution. However, based on the empirical results, **we still argue that it achieves reasonable results for non-Gaussian distributions**. We respectfully disagree with the reviewer saying the method is only applicable to the Gaussian distribution. **We have demonstrated applicability to Beta and Gamma distributions** also, and we now mention this in the second paragraph in section 2:
> >”We propose parameter updates for a Beta and Gamma Proposal Sampler, see Appendix I.”
>
> While we in principle agree on diagonal that the diagonal covariance assumption is crude, the reviewer is well aware that **most methods make the assumption that parameters are independent when dealing with large models. Generally, this approximation is considered both acceptable and necessary because of the huge computation and memory cost.** We want to stress that even though the proposal is a diagonal Gaussian, the resulting posterior approximation is not constrained to a diagonal Gaussian. This is now stated more clearly regarding the proposal in paragraph after equation 7:
> > “For theta in R^d, the parameters are considered independent, making the proposal a diagonal multivariate Gaussian”.
>
> It may be open for future research to incorporate a covariance matrix.
>
> We specify a perfect sampler in our revision. In “Related work” paragraph 2 it now reads:
> > “[...] a perfect sampler, meaning that it must produce independent samples from the true distribution,”
>
>  and in paragraph 6 on page 2:
> >“ A perfect sampler q generates a sample \theta with the exact probability of f (θ) (Propp & Wilson, 1996).”
>
>
> Additionally, we have modified our claims in the first and second paragraph in the concluding section:
> >“§1[...] A-GPS excels in unimodal scenarios, demonstrating fast parameter updates and high-quality sampling. ”
>
> >“§2 [...] Compared to established methods like HMC and Langevin dynamics, a-GPS showcases superior computational speed while maintaining a substantial effective sample size, especially within modes.” Showcasing that the allure of exactness is predominantly within modes. We are unfortunately not allowed to make any changes to the abstract but will add this in a potential camera ready version.
>
> **Comment 2:**
> *The methodology of the proposed method is confusing. The algorithm does not have a component to encourage exploring multiple modes. It is unclear to me how the method manages to find diverse modes.*
>
> **Answer 2:**
> We apologize for the confusion regarding our method. **Our algorithm navigates complex landscapes in deep learning by approximating the mode of a given point, drawing random samples, and dynamically adjusting to overlapping modes. This adaptability enhances its ability to capture uncertainty in multimodal distributions, offering a flexible approach compared to traditional methods.** The text has been revised to make this methodology clearer, see paragraph the first  paragraph under “Adaptive Proposal Sampling” which now reads:
> >“This mode-seeking sampler adapts q to align with the score function of a posterior mode. In the case of overlapping modes, APS adjusts to a new mode when it draws a sample belonging to a new mode“
>
>
> **Comment 3:**
> *Algorithm 1 seems to find a Gaussian distribution to approximate the target distribution. How is it different from variational inference? What are the advantages?*
>
> **Answer 3:**
> We are grateful for this important question. The proposed method is different from variational inference (VI): **contrary to VI, our method’s posterior approximation is not restricted by the chosen family of q**.

---

> ### Author Response · Authors · 2023-11-23
> **Response to zCTz**
>
> **Comment 4:**
> *Why does the proposed method require a pretrained solution, theta_MAP? Will it work if training from scratch?*
>
> **Answer 4:**
> We appreciate the encouragement to further clarify this in our manuscript. **The proposed method does not require a pre-trained solution.** However, there is little interest in sampling the density of a randomly initialized network because the samples do not correspond to a desirable model. We kindly remark that it does not optimize a loss, it just uses the gradient from the loss in order to greedily estimate the underlying distribution, then draws a random sample from this distribution. This is now expanded on within the manuscript, paragraph 2 under “ImageNet Results” now reads:
> > “We do emphasize that a-GPS does not optimize a loss to achieve good performance, it draws a random sample from the approximation of the landscape.”
>
>
> **Comment 5:**
> *I do not follow the reason for introducing the variance limit lambda. Why does the method need it?*
>
> **Answer 5:**
> We thank the reviewer for this question. **The variance limit $\lambda$ was introduced to prevent excessive induced noise especially around flat areas like saddle points.** We rephrased the description of a variance limit (in the last paragraph of sec 2.1) in the main text such that it will appear before the pseudo-algorithm (sec 2.3.1, page 5), it now reads
> >“For stability and efficiency, we introduce an upper variance limit $\lambda$. This hyperparameter limits the potential risk of overestimating the underlying variance around saddle points or other flat areas, or to ensure that samples remain sufficiently close to high density area (see Appendix E, for more details). ”
>
> **Comment 6:**
> *The experimental setups and results are confusing. It is unclear if the authors also use a pre-trained solution for the baseline NUTS in S3.1. If not, then it is unfair to claim faster convergence of the proposed method than NUTS. Besides, given that the method uses a pre-trained solution, it is unsurprising that “We found that a-GPS converges so fast that a burn-in period was unnecessary”. For the time comparison, it is unclear if the authors include pre-training time.*
>
> **Answer 6:**
> We thank the reviewer for prompting us to provide clarification on this. For the simulated experiments, **neither aGPS nor NUTS (or Langevin in the updated version) utilized pre-trained solutions, to ensure an unbiased comparison, all initializations were randomly drawn from Normal(0,1) across the 100 replications.** We have updated Section 3.1 to state more explicitly that in all simulations, $\theta_1$ was initialized from N(0,1) for both methods (last paragraph on page 5 and third paragraph page 6). The reviewer correctly noted that NUTS requires a burn-in time to tune its hyperparameters, while aGPS does not. We rewrote it to
> >“We found that a-GPS converges towards a perfect sampler in just a few steps so a dedicated burn-in period was unnecessary.”
>
> It is correct that the timing includes any burn-in for the simulated examples. To avoid confusion we updated the Table 1 caption to:
> >“We report the average time (in seconds) for (burn-in + 1000 samples) and Effective Sample Size (ESS) for 1000 samples  over 100 replications.”
>
> **Comment 7:**
> *For deep learning experiments, it will be better to include MCMC baselines, e.g. Zhang et al, as the proposed method belongs to MCMC methods. To show the samples are from diverse modes, the authors can visualize weight space and function space, similar to those in Zhang et al.*
>
> **Answer 7:**
> We regarded cyclical SG-MCMC as an extension of (SG-)MCMC baselines like HMC and Langevin dynamics; and thus they were not included  in our experiments. It could be interesting to visualize the weight and function space, but it is out of our current scope. Like most extensions (i.e., cyclical schedule and temperature scaling) this may also be applied to our framework.

---

> ### Author Response · Authors · 2023-11-23
> **Response to zCTz**
>
> **Question 1:**
> *Why is LA’s inference time even less than MAP? Why is the proposed method’s inference time less than SWAG? Does the proposed method use Bayesian model averaging during inference?*
>
> **Answer 1:**
> The inference time for LA being faster than MAP on CIFAR10 is now commented on in the first paragraph under “CIFAR 10 and CIFAR100 Results”, it now reads:
> >“ We report LA to be faster than MAP and hypothesize that this is due to the optimisation of LA happening just before inference, thus all or some of the data may be in memory.”
>
> The inference time of SWAG is likely affected by batch-norm updates, a topic we refer to for details in Appendix H :
> >”For SWAG Maddox et al. (2019) suggest to continually update the batch norm parameters with 10% of the training data for each sample.”
>
> All models employing multiple forward passes (SWAG, a-GPS, Dropout) utilize Bayesian model averaging for computing metrics such as Acc, ECE, and NLL, this is now explicitly stated in the first paragraph under “Deep Learning Experiments” :
> >“[...] bottom section in tables includes methods that perform a Bayesian model average with multiple forward passes during inference.”

---

### Official Review · Reviewer_RZPX · 2023-10-30

**Soundness:** 3 good
**Presentation:** 2 fair
**Contribution:** 2 fair
**Rating:** 3
**Confidence:** 4

**Summary:**

The paper proposes a sampler that samples weights via traversing the loss landscape of a pre-trained deep neural network via a series of normal distributions. The approach is evaluated on a series of classification and out-of-distribution detection tasks.

**Strengths:**

The paper proposes a sampler that samples weights via traversing the loss landscape of a pre-trained deep neural network via a series of normal distributions. The approach is evaluated on a series of classification and out-of-distribution detection tasks.

**Weaknesses:**

- The main weakness of the paper is in the experimental evaluation. The experiments show convincingly that the proposal works with several architectures and several classification data sets (no regression tasks were evaluated). What it does not show is that it works better than its baselines, i.e., why should it be used instead of SWAG, or SGD-MC? E.g., SGD-MC almost always outperforms it (it is missing from Table 4, but the results in Table 13, show that it clearly performs better), except for the strange behavior in Table 6.


- The presentation of the paper is rather sub-optimal. E.g.,
    - parameters such as $c$ and $\lambda$ appear in the text long before they are even introduced, if at all. The important $\lambda$, e.g., only is further detailed in Algorithm 1.
    - The writing contains a lot of typos, e.g., for the first paragraph on the second page
        - "full-gradient MCMC similar **to** SG-MCMC"
        - "SGLD **has** fast computations but **suffers** form inefficient explorations"
        - "Previous **works** on state dependent"
    - Dropout's absence in most of the results is not explained in the main text but only appears in the one table where it is present rather than absent
    - The writing is somewhat repetitive
    - The reference list is full of arxiv preprints instead of the actual publications
    - Table 4 contains wrong highlights in two columns (ECE and NLL), the same is true for several tables in the appendix.
    - On the positive side, however, other details, like definitions of performance metrics are highlighted prominently

### Minor
- SGD-MC is mentioned in the text for Table 4 but not in the actual results
- LA is missing in Table 3 without an explanation
- Sec 2.1: "the loss function, ..., typically cross-entropy is interpreted as the negative log-likelihood". Cross-entropy is typical for classification tasks, but not for any other tasks. And in this case, it is not just interpreted as a negative log-likelihood, _it is_ the negative of a categorical distribution.
- For the posterior in  (15). A Gaussian prior is $\exp(-||\theta||)$, similarly for the loss factor. This directly provides you with (17) instead of having to redefine anything.
- Sec 3.2.2 "separated by high loss area". As Draxler et al. (2018) and Garipos et al. (2018) show there are a lot of paths of similar loss between a lot of maxima instead of a clear separation. (These motivated the SWA baseline of the present work)



_____
Draxler et al., _Essentially no Barriers in Neural Network Energy Landscape_, ICML 2018
Garipov et al., _Loss Surfaces, Mode Connectivity, and Fast Ensembling of DNNs_, NeurIPS 2018

**Questions:**

- The conclusion only discusses a-GPS' performance with respect to SWAG and Laplace. Can the authors additionally provide a deeper discussion on their relation to SGD-MC and in general summarize why their approach should be picked instead of these established baselines?
- SGLD is mentioned in the related work, but never used in the experiments. Can the authors comment on this lack of comparison? Especially since they cite Izmailov et al. (2021) who showed good results for this approach.
- A lot of approaches and networks diverged or failed otherwise throughout the experiments. Can the authors give further details? E.g., it seems rather strange that a simple model such as VGG should diverge on a straight-forward classification task such as CIFAR100.
- The method was only tested on classification tasks. What about regression problems? Do the authors expect a similar performance?
- How is the split in CIFAR10 and CIFAR 100 in 5/50 classes decided? _(Apologies if I missed it somewhere in the appendix)_

---

> ### Author Response · Authors · 2023-11-23
> **Response to RZPX**
>
> **Comment 1:** *“The experiments show convincingly that the proposal works with several architectures and several classification data sets, but no regression tasks were evaluated, and it does not show is that it works better than its baselines (SWAG, or SGD-MC) E.g., SGD-MC almost always outperforms it, except for the strange behavior in Table 6.*
>
> **Answer 1:**
> We thank the reviewer for the effort put into the thorough and constructive feedback, and for recognizing the success of the method across various architectures and classification datasets.
>
>
> We agree with the reviewer that the method does not surpass the baselines on standard performance metrics. However, we still believe the method is a substantial contribution as it is **a novel sampler that quickly produces a posterior approximation for any model, it can be multimodal, with high effective sample size within modes**. In contrast to our model, SWA, SWAG and LA may be limited by their Gaussian posterior approximation, while SG-MCMC methods may be limited by their inefficient exploration. This has now been added both in the introduction results, discussion and conclusion, see the last paragraph on page 1, paragraph 1, 3 and 4 on page 2, paragraph 3 on page 5 and paragraph 3 on page 8.
>
> We have fully revised the conclusion to emphasize the advantages of the model compared to all the methods presented in our results. The concluding paragraph 5, page 8, now reads:
> >“[...] Compared to established methods like HMC and Langevin dynamics, a-GPS showcases superior computational speed while maintaining a substantial effective sample size, especially within modes. In deep learning contexts, a-GPS achieves comparable results to SWA, SWAG, and SGD-MC with significantly reduced training time [...]. During inference, a-GPS demonstrates linear computational complexity with the number of samples. Effectively scaling to large datasets, unlike SWA and SWAG, it maintains a balanced trade-off between computational speed and model exploration.”
>
> Regression is an interesting problem to investigate in the future. We regard it as outside of the current scope of this paper.
>
> **Even if statistical novel models don't outperform existing standards, their competitiveness makes them worthy of exploration and publication. By offering unique approaches, these models contribute valuable insights to the field.** We agree with the reviewer that this should have been explicitly stated. Paragraph 2 on page 9 now reads:
> >"While not surpassing standards universally, the model showcases competitiveness as a novel sampler, swiftly generating a posterior approximation for any model. Its multimodal capability, coupled with a high effective sample size within modes, underscores the need for further exploration and elaboration.”
>
> The reviewer pointed out that the explanation for the data in Table 6 was unclear, describing it as 'strange behavior.' In response, **we've provided a clearer and more detailed explanation** for the data presented in Table 6. Last paragraph, page 8 now reads:
> >“In Table 6, we observe undesirable behavior from SGD-MC and LA as they exhibit higher uncertainty for IND than OOD. SGD-MC relies on the inherent noise present in the data, and as a result, fluctuations in the training set may lead to inconsistent uncertainty estimates. When the assumptions of LA do not align with the true distribution, especially in complex and multimodal scenarios, LA fails to accurately capture the underlying uncertainty. In the case of our experiments, both SGD-MC and LA's inconsistency in uncertainty estimates for in-distribution and out-of-distribution data suggests a failure to adapt to the true data distribution. Once again, a-GPS demonstrates greater certainty about IND and relatively higher uncertainty for OOD, though SWA exhibits the highest uncertainty for OOD.”
> We have also added SGD-MC in table 4.

---

> > ### Author Response · Authors · 2023-11-23
> > **Response to RZPX**
> >
> > **Comment 2:**
> > *“The presentation of the paper is rather sub-optimal. E.g., parameters such as “c” and “lambda appear in the text long before they are even introduced; the writing contains a lot of typos.. Dropout's absence in most of the results is not explained in the main text but only appears in the one table where it is present rather than absent; The writing is somewhat repetitive; Reference list is full of arxiv; Table 4 contains wrong highlights, this also applies to other tables.*
> >
> > **Answer 2:**
> > We have carefully updated the paper and hope this revision better presents our contribution.
> >
> > Firstly, we appreciate the feedback regarding the clarity and organization of the paper, and **we have enhanced organization, spelling and language**, for improved readability. The list of references has also been carefully updated, and highlight-mistakes in tables have been corrected (see table 4, 11, 12 & 13).
> >
> > In the revised version **we have addressed the role of the constant “c” with more clarity regarding its role as a constant**;  the paragraph after equation 1.We also rephrased the description of a variance limit (in the last paragraph of sec 2.1) in the main text such that it will appear before the pseudo-algorithm (sec 2.3.1, page 5), it now reads
> > >“For stability and efficiency, we introduce an upper variance limit $\lambda$. This hyperparameter limits the potential risk of overestimating the underlying variance around saddle points or other flat areas (see Appendix E, for more details). ”
> >
> > In the revised version paragraph 2 page 2 we state that
> >  >“ Monte Carlo dropout (MC-dropout) [...] requires an architecture with dropout layers.”.
> > The first paragraph in “Results and Discussion” has also been updated with a note that:
> >  >“MC-dropout (Gal & Ghahramani, 2016) (Note: VGG-16 is the only architecture with dropout nodes)”.
> >  Finally,
> > >“However, MC-Dropout is limited to dropout architectures.” is mentioned in the last paragraph page 6.
> >
> > **Comment 3 (MINOR):** *“SGD-MC is not in the table 4 results; LA missing in table 3; Sec 2.1:*
> > **Answer 3:** We thank the reviewer for noticing this. The missing **results in table 3-4 have now been included.**
> >
> > **Comment 4 (MINOR):** *Sec 2.1: “ the [...] “ cross entropy is the negative of a categorical distribution. For the posterior in (15). A Gaussian prior is exp(-||theta||), similarly for the loss factor. This directly provides you with (17) instead of having to redefine anything. Sec 3.2.2 "separated by high loss area". As Draxler et al. (2019) and Garipos et al. (2018) show there are a lot of paths of similar loss between a lot of maxima instead of a clear separation. (These motivated the SWA baseline of the present work)*
> >
> > **Answer 4:**
> > We thank the reviewer for pointing this out. **We removed the mention of cross entropy** because it is applicable for any loss function. We moved the whole passage about how the posterior relates to the loss and regularization to the Appendix A. The updated version is more general, the last paragraph of Appendix A now reads:
> >  >”In the context of deep learning, the objective is to minimize a loss function with respect to the parameters given the data …”
> > We have corrected the text according to your point of exp(-||theta||), making it a more succinct paper.
> >
> > The reviewer highlighted two great papers regarding the deep learning loss landscape. These works provide a similar notion of the loss landscape that motivated our reasoning of using a-GPS for deep learning. **The reason we suggest that a-GPS explored modes “separated by high loss area” is because the SWA (mean) and SWAG (mean+cov) based on the samples obtained by a-GPS-4 resulted in an accuracy of 39% and 45%, respectively while a-GPS-4 stayed at 91%.** The last sentence in the last paragraph on page 6 now reads:
> > >“this suggests that a-GPS-4 has traversed different modes separated by relatively high loss (low accuracy) areas.”

---

> ### Author Response · Authors · 2023-11-23
> **Response to RZPX**
>
> ### Questions
> **Question 1:**
> *The conclusion only discusses a-GPS' performance with respect to SWAG and Laplace. Can the authors additionally provide a deeper discussion on their relation to SGD-MC and in general summarize why their approach should be picked instead of these established baselines?*
>
> **Answer 1:**
> This is an excellent suggestion. **We have updated the conclusion and discussion to include the models HMC, Langevin dynamics, LA, Dropout, SWA, SWAG and SGD-MC.** Paragraph 2 and 3 on page 9 now reads:
> >"§2 A-GPS exhibits computational efficiency during training, rivaling SGD in speed, making it a feasible option for time-sensitive tasks. Compared to established methods like HMC and Langevin dynamics, a-GPS showcases superior computational speed while maintaining a substantial effective sample size, especially within modes. In deep learning contexts, a-GPS achieves comparable results to SWA, SWAG, and SGD-MC with significantly reduced training time, suggesting notable efficiency gains. During inference, a-GPS demonstrates linear computational complexity with the number of samples. Effectively scaling to large datasets, unlike SWA and SWAG, it maintains a balanced trade-off between computational speed and model exploration.
>
> >§3 The observed behaviors in methods like SGD-MC and LA emphasize the need for a nuanced and adaptive approach to uncertainty modeling. Unlike a-GPS, these conventional methods may lack the flexibility required to navigate diverse data distributions and could oversimplify complex structures inherent in real-world datasets. A-GPS, with its adaptability and exploration capabilities, emerges as a promising alternative that aligns well with the intricacies of uncertainty in various scenarios
>
>
> **Question 2:**
> SGLD is mentioned in the related work, but never used in the experiments. Can the authors comment on this lack of comparison? Especially since they cite Izmailov et al. (2021) who showed good results for this approach.
>
> **Answer 2:**
> We are grateful for the opportunity to clarify this. **We did not compare with SGLD or SGHMC on deep learning problems as LA, SWA and SWAG have achieved equal or better performance (Izmailov et al., 2021)**. In addition, for the task of converting an existing pretrained deep learning model into a Bayesian network, **the methods we compared did not require an extensive burn-in/training time and extensive hyperparameter search**. We have clarified this at the end of the last paragraph in page 1: >“However, SG-MCMC methods rely on inefficient proposal dynamics that generates few effective samples per epoch, even from a warm-start, making it expensive for deep learning problems.”. Finally, we included Langevin dynamics in the simulated experiments. Table 1 now reads
> | Method      | Unimodal ESS | Unimodal Time (s) | Mixture ESS | Mixture Time (s) |
> |-------------|--------------|-------------------|-------------|-------------------|
> | **a-GPS**    | 954          | 0.4               | 70.2        | 2                 |
> | **HMC**      | 364          | 4.6               | 105.2       | 30                |
> | **Langevin** | 74           | 2.3               | 36.8        | 4.9               |
>
>
>
> **Question 3:**
> *A lot of approaches and networks diverged or failed otherwise throughout the experiments. Can the authors give further details? E.g., it seems rather strange that a simple model such as VGG should diverge on a straight-forward classification task such as CIFAR100.*
>
> **Answer 3:**
> In our experiments, **the VGG-16 diverged with the Laplace method because it requires O(n^2) storage of the last layer parameters, which resulted in out-of-memory errors. We have addressed this in the revised version** in Appendix G.1:
> > “Due to the O(n^2) memory requirement of LA we experienced out-of-memory errors for the VGG-16 architecture on CIFAR100, we had to deviate from the standard recommendations, by changing prediction type to 'nn' and link approximation to 'mc'.”
>
> **Question 4:**
> *The method was only tested on classification tasks. What about regression problems? Do the authors expect a similar performance?*
>
> **Answer 4:**
> This is an interesting problem to investigate in the future. We regard it as outside of the current scope of this paper.
>
> **Question 5:**
> *How is the split in CIFAR10 and CIFAR 100 in 5/50 classes decided? (Apologies if I missed it somewhere in the appendix)*
>
> **Answer 5:**
> We agree with the reviewer that this was not clear. For clarity we added
> >“[..]the data was split class-wise into two equally sized groups; (0, 1, 2, 8, 9) and (3, 4, 5, 6, 7) for CIFAR10 following (Maddox et al., 2019), and 0->49 and 50->99 for CIFAR100”
>
>  to the first paragraph in the “Out-of-Distribution” section.

---

### Meta-Review · Area_Chair_NCqg · 2023-12-05

**Metareview:**

This submission was reviewed by four reviewers. Even if the topic itself is a good fit and interesting for ICLR, all reviewers found multiple issues with this submission. I recommend taking the reviewer input into account before considering resubmitting this work to another venue.

**Justification For Why Not Higher Score:**

Multiple issues with the submission. All reviewers (and AC) agreed.

**Justification For Why Not Lower Score:**

N/A

---

### Decision · Program_Chairs · 2024-01-16

Reject